# `AlphaDecay`: Module-wise Weight Decay for Heavy-Tailed Balancing in LLMs

**Di He**[1,2,3], **Songjun Tu**[2,3], **Ajay Jaiswal**[4], **Li Shen**[5], **Ganzhao Yuan**[6],
**Shiwei Liu**[7], **Lu Yin**[8] †

[1]Shenzhen Institutes of Advanced Technology, Chinese Academy of Sciences
[2]Peng Cheng Laboratory    [3]University of Chinese Academy of Sciences
[4]University of Texas at Austin    [5]Shenzhen Campus of Sun Yat-sen University
[6]Shenzhen University of Advanced Technology
[7]University of Oxford    [8]University of Surrey
l.yin@surrey.ac.uk

## Abstract

Weight decay is a standard regularization technique for training large language models (LLMs). While it is common to assign a uniform decay rate to every layer, this approach overlooks the structural diversity of LLMs and the varying spectral properties across modules. In this paper, we introduce `AlphaDecay`, a simple yet effective method that adaptively assigns different weight decay strengths to each module of an LLM. Our approach is guided by Heavy-Tailed Self-Regularization (HT-SR) theory, which analyzes the empirical spectral density (ESD) of weight correlation matrices to quantify "heavy-tailedness." Modules exhibiting more pronounced heavy-tailed ESDs, reflecting stronger feature learning, are assigned weaker decay, while modules with lighter-tailed spectra receive stronger decay. Our method leverages tailored weight decay assignments to balance the module-wise differences in spectral properties, leading to improved performance. Extensive pre-training tasks with various model sizes from 60M to 1B demonstrate that `AlphaDecay` achieves better perplexity and generalization than conventional uniform decay and other adaptive decay baselines. The code is available at https://github.com/heducas/AlphaDecay.

## 1 Introduction

Large language models (LLMs) have emerged as a core technology in artificial intelligence, with extensive applications in chatbots, content generation, code synthesis, and other domains, significantly enhancing the efficiency and user experience of human-computer interaction [5; 28; 23; 11]. However, the formidable capabilities of these models rely on massive pre-training datasets and substantial computational resources, rendering the training process fraught with challenges [44; 2]. Persistent research challenges include, but are not limited to, the efficient optimization of ultra-large-scale parameters and the trade-off between training costs and model performance.

Weight decay, one of the most widely used regularization techniques for training well-generalized deep neural networks [25; 42; 13], critically influences the convergence and performance of state-of-the-art machine learning algorithms when properly configured. Extensive prior studies [21; 32; 35] have demonstrated its pivotal role in enhancing model generalization from diverse theoretical and empirical perspectives. Recent work [20; 8] further highlights its importance in improving optimizer stability and efficacy during the training of LLMs.

---

†Corresponding author.

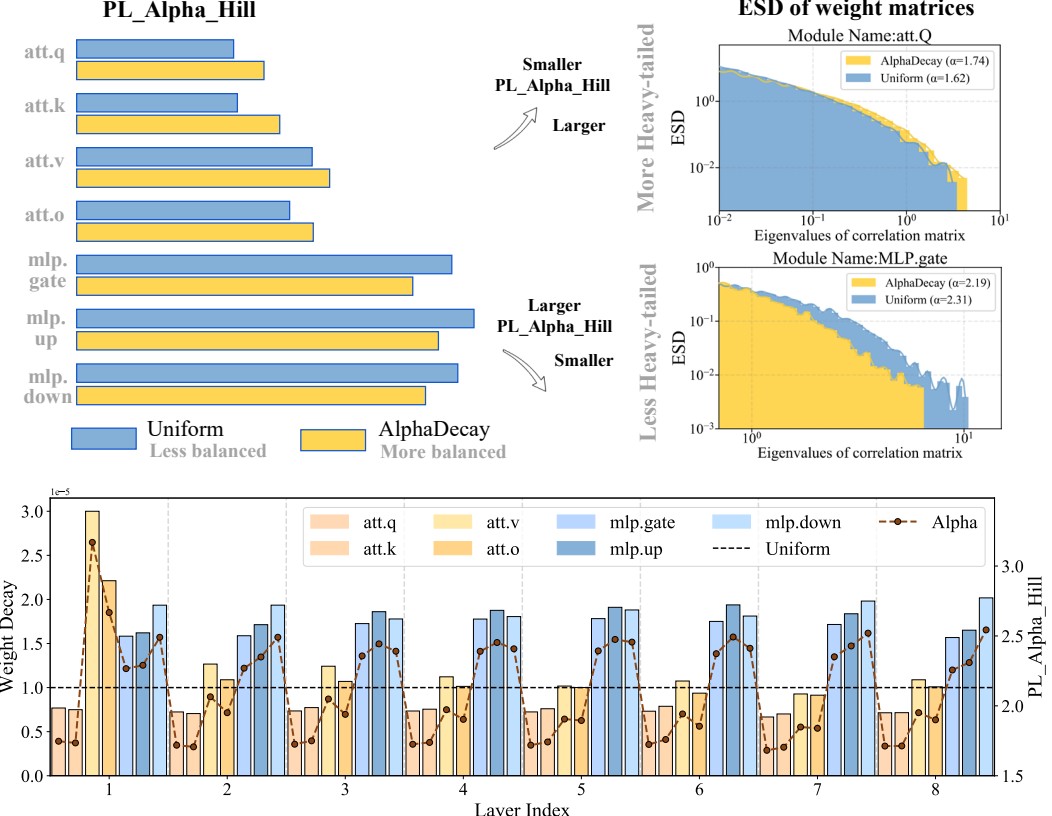

Figure 1: **Module-wise Balance and `AlphaDecay` weight decay schedule.** (a) Employing PL fitting to derive module-wise `PL_Alpha_Hill` values (see formula (2)), `AlphaDecay` achieves module-wise balance by increasing the lower values (e.g., `att.Q` and `att.K`, more heavy-tailed) while decreasing the higher values (e.g., MLP components, less heavy-tailed). (b) Given the imbalanced module-wise `PL_Alpha_Hill` of LLaMa-60M, `AlphaDecay` assigns lower weight decay to modules with lower `PL_Alpha_Hill`.

The prevailing approach to weight decay assigns a globally fixed value per epoch across optimizers—including SGD [39], Adam [18], and their variants [47; 36]—where all model layers share an identical decay coefficient. However, given the scaling parameter counts and architectural complexity of modern LLMs, such a uniform weight decay scheme fails to capture their intricate structural properties, making this conventional practice increasingly suboptimal. Notably, recent work has begun investigating dynamic weight decay adaptation [16; 30; 12; 42] to address this limitation. [12] observes that fixed-hyperparameter weight decay fails to balance robustness and accuracy in adversarial training, causing robust overfitting. They propose Adaptive Weight Decay (`AWD`) to dynamically adjust decay strength via classification and regularization loss gradients, automatically enhancing robustness and adversarial performance without extra data.

Notably, prior studies on dynamic weight decay adaptation were exclusively designed for architectures like ResNet18/34/50 [14], VGG [34], and DenseNet [15], employing time-wise modulation (i.e., uniform decay values across all layers at each timestep) while maintaining layer-wise uniformity. This approach is reasonable for parameter-efficient, structurally simple models (e.g., ResNets) where inter-layer feature distinctions are less pronounced. However,

> *Does there exist a better weight decay configuration for LLMs?*

Three reasons behoove us to pose the above research question: First, the prevailing consensus holds that certain transformer components exhibit greater functional importance than others [40; 4; 44; 26], necessitating differentiated weight decay treatment. Second, weight decay manifests fundamentally distinct roles in over-trained regimes (e.g., ResNets) versus under-trained regimes (e.g., LLMs) [8].

Most notably, existing research demonstrates that improper weight decay configuration for LLMs may adversely affect model performance [3; 19; 42; 33; 17; 9]. Our main contributions are as follows:

❶ We identify substantial variation in the spectral properties of module-wise ESD (see figure 2), and show that these inconsistencies are a core reason for degraded model performance, as evidenced by figure 4.

❷ We propose a module-wise weight decay scheduling strategy `AlphaDecay` to ensure spectral alignment across modules (see figure 1), thereby enforcing consistency in spectral properties and achieving improved training performance (see figure 3).

❸ Extensive experiments spanning models from 60M to 1B parameters show that the proposed approach, `AlphaDecay`, consistently outperforms the `Uniform` baseline as well as adaptive methods such as `AWD` [12] and `AdaDecay` [30] (see table 2). These results highlight the critical role of module-wise balance in achieving state-of-the-art performance in LLMs.

Overall, our research provides an unrecognized perspective on optimizer, revealing the critical yet overlooked role of module-wise weight decay in LLM training. This novel insight can be readily applied to all state-of-the-art optimizers and training methods, effectively enhancing their performance without structural modifications.

## 2  Related Work

**Weight decay in LLM training**. Weight decay is a widely adopted technique for training deep networks, spanning applications from image classification to LLMs [21]. In the context of GPT-3 training, [5] recommended incorporating weight decay primarily for its mild regularization benefits. [20] showed that weight decay promotes optimizer equilibrium in scale-invariant systems. Recent studies have provided deeper insights into weight decay's role in LLM training. [38; 8; 37] challenged the conventional view of weight decay's generalization benefits for LLMs, and instead highlighting its critical function in reducing training loss and enhancing stability during under-training through the lens of Effective Learning Rate. Building on these findings, [3; 19] established a connection between $l_2$ regularization and spectral norms, discovering that weight decay induces low-rank attention layers. Their work further showed that employing different weight decay values for attention and MLP modules, carefully tuned via grid search, can significantly improve training outcomes. Our work presents the first formal analysis of non-uniform module-wise weight decay in LLM training, demonstrating its effectiveness through comprehensive empirical validation.

**Dynamic weight decay**. While uniform weight decay is commonly used for model training, a line of work employs gradnorm to adaptively determine weight decay settings. [16] analyzed gradient descent with weight decay, finding that backpropagated gradients scale with upstream weights while weight decay scales with each layer's own weights. This mismatch in scaling causes layer-wise overfitting or underfitting, leading them to propose using the gradient-to-decay magnitude ratio as a layer-wise coefficient. [30] enhanced this approach by normalizing gradients and applying a scaled sigmoid to compute the coefficient. Similarly, [12] used the ratio of gradient norms to parameter norms. [42] showed weight decay amplifies late-stage gradient norms, harming convergence. Their solution, AdamS, penalizes large gradients and outperforms both Adam and AdamW. Another line of research [3; 10; 41] revealed that weight decay induces low-rank layer structures. [19] further showed that applying distinct weight decay values to attention and MLP modules, meticulously tuned via grid search, can substantially enhance training outcomes. Building upon these foundations, our work advances this direction by introducing the first weight decay scheduling framework for LLMs.

**Heavy-tail self-regularization**. HT-SR Theory examines the ESD of weight matrices and identifies its relationship with training quality based on principles from statistical physics and random matrix theory [7]. HT-SR Theory posits that well-trained neural networks exhibit strong correlations in their weights, manifesting as heavy-tailed structures in the ESD of each layer's weight matrics [27; 29]. Recently, HT-SR has been applied to model selection [27; 29; 43], module-wise adaptive training [46], and LLM pruning [26], demonstrating its efficacy in estimating model and layer quality. However, no prior work has explored HT-SR theory in the context of weight decay configuration. Our work draws inspiration from HT-SR theory and introduces a novel technique that leverages ESD structures to guide weight decay settings.

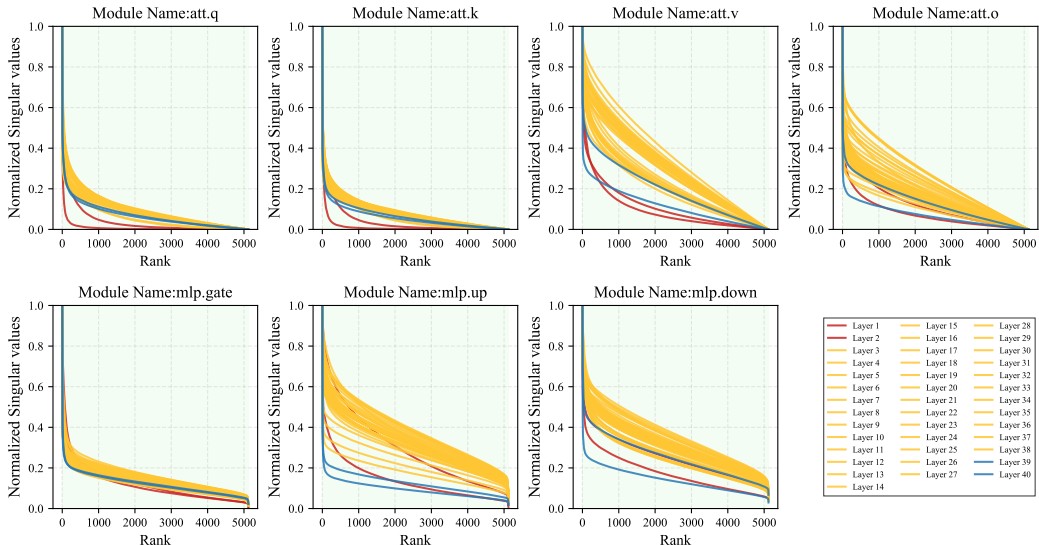

Figure 2: Visualization of singular values from weight matrices in each layer of the pretrained LLaMa-2-13b-hf model. For all 40 transformer layers, the plots show the sorted distribution of 5120 singular values per layer.

# 3 Methodology

In this section, we first present the rationale motivating our study, emphasizing the heavy-tailed singular value spectra exhibited by different modules of LLMs. We then revisit HT-SR theory and introduce key HT-SR metrics that support our analysis. Finally, we examine the `AlphaDecay` algorithm, which leverages "shape metrics" derived from HT-SR theory and exhibits significant improvements in LLM pretraining tasks.

## 3.1 Rationale

Different modules in LLMs exhibit diverse spectral properties, particularly in the distribution of their singular values. Figure 2 visualize the normalized singular value spectra of the weight matrices for each module type (`att.q`, `att.k`, `att.v`, `att.o`, `mlp.gate`, `mlp.up`, `mlp.down`) across all 40 transformer layers in the pretrained LLaMa-2-13b-hf model. Notably, substantial variability is observed in the heavy-tailedness of the singular value distributions: the attention-related modules (`att.q` and `att.k`) consistently show heavier tails, while the MLP modules (`mlp.gate`, mlp.up, mlp.down) exhibit lighter tails.

This phenomenon has been extensively studied within heavy-tailed random matrix theory. Specifically, heavier tails in the singular value spectra reflect greater anisotropy, with much of the module's representational power concentrated in a few leading principal components—a feature especially pronounced in attention-related modules (`att.q`, `att.k`). In contrast, the lighter-tailed spectra observed in MLP modules (`mlp.gate`, `mlp.up`, `mlp.down`) exhibit a more uniform distribution across components. These observations suggest that different modules may benefit from tailored regularization strengths to achieve optimal performance, as attention modules could be more disrupted by excessive regularization, while MLP modules may tolerate stronger regularization.

## 3.2 HT-SR Theory

The HT-SR theory provides a principled framework for analyzing the empirical spectral distribution (ESD) of neural network weight matrices. Empirical evidence suggests that well-trained models exhibit more pronounced heavy-tailed ESDs, which reflect higher training quality. Building on this theoretical foundation, our method leverages the HT-SR metric to quantify spectral tail heaviness, assigning lower weight decay to heavily-tailed modules (e.g., `att.q`, `att.k`) and higher weight decay to less heavy-tailed ones (e.g., MLP components), thereby aligning with spectral characteristics to potentially improve generalization and model performance (see figure 1). The degree of heavy-

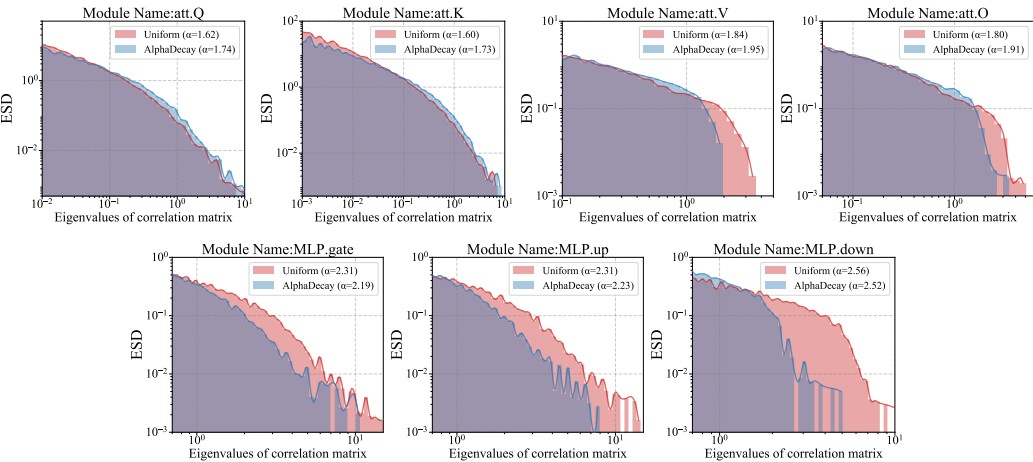

Figure 3: Comparison of ESD distributions across modules of LLaMa-135M under different training methods (`AlphaDecay`: Perplexity=22.55 vs. `Uniform`: Perplexity=23.14). Attention-related modules (e.g., `att.q`, `att.k`) exhibit notably heavier spectral tails in contrast to MLP-associated modules. Our method systematically balances the heavy-tailed properties across modules by appropriately configuring module-wise weight decay, thereby enhancing overall model performance.

tailedness is quantitatively assessed by fitting a power law (PL) to the ESD, using the resulting PL exponent ($\alpha$) as a metric.

Given a network with $L$ modules and weight matrices $\{\mathbf{W_l}\}_{l=1}^L$ of shape $n \times m$ ($n \leq m$), we compute the ESD by obtaining the eigenvalues of the correlation matrix $\mathbf{X}_l = \mathbf{W}_l^\top \mathbf{W}_l$ for each module. The power law fit for the ESD takes the form:

$$p(\lambda) \propto \lambda^{-\alpha}, \lambda_{\min} < \lambda < \lambda_{\max} \tag{1}$$

where $p(\lambda)$ denotes the density of eigenvalues $\lambda$ within the specified range. The PL exponent, $\alpha$, serves as a proxy for the degree of heavy-tailedness.

To estimate $\alpha$, we use the Hill estimator [46; 24]. For a given module's eigenvalues $\{\lambda_i\}_{i=1}^n$ (sorted in ascending order), the Hill estimator is given by:

$$\texttt{PL\_Alpha\_Hill} = 1 + \frac{k}{\sum_{i=1}^k ln \frac{\lambda_{n-i+1}}{\lambda_{n-k}}} \tag{2}$$

where $k$ controls the lower cutoff for PL fitting. In our experiments, we fix $k = \frac{n}{2}$, i.e., we estimate the slope using the largest half of the eigenvalues.

`PL_Alpha_Hill` is a key spectral descriptor for analyzing model performance. Related works [46; 24] suggest that lower `PL_Alpha_Hill` values indicate "overtrained" layers (compared to other layers in the model), while higher values indicate "undertrained" layers. An important conclusion is that a more uniform distribution of `PL_Alpha_Hill` across layers reflects more balanced training quality, leading to better overall model quality. While these findings highlight the importance of layer-wise training balance, our work emphasizes a complementary perspective:

> *Does module-wise balance matter for model performance*?

We empirically demonstrate that promoting uniformity in `PL_Alpha_Hill` across modules (e.g., attention and MLP components) can further enhance overall model quality (see figure 3).

### 3.3 `AlphaDecay`

Building on the observed spectral diversity across modules, we introduce `AlphaDecay`, a simple yet effective module-wise weight decay scheduling algorithm. `AlphaDecay` first calculates the `PL_Alpha_Hill` values for all modules, and then assign larger weight decay to modules with higher

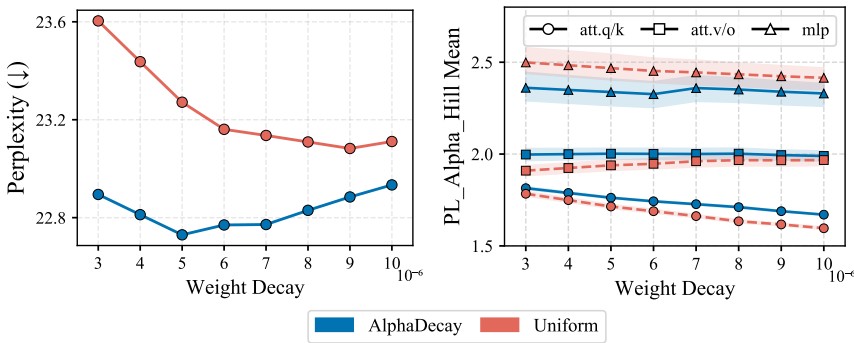

Figure 4: Comparison of perplexity and module-wise `PL_Alpha_Hill` values of LLaMa-135M under varying weight decay settings; For each group, `att.q/k` shows the mean `PL_Alpha_Hill` of `att.q` and `att.k`; `att.v/o` shows the mean for `att.v` and `att.o`; `mlp` is the mean of `mlp.gate`, `mlp.up`, and `mlp.down`. Shaded areas indicate the range between the maximum and minimum values within each group.

`PL_Alpha_Hill` values, while assigning smaller weight decay to those with lower `PL_Alpha_Hill` values. This strategy is designed to promote module-wise `PL_Alpha_Hill` balance, thus leading to better overall model performance. We provide the details of `AlphaDecay` in Algorithm 1. The assignment function is given by:

$$f_t(i) = \eta \cdot \left( \frac{\alpha_t^i - \alpha_t^{\min}}{\alpha_t^{\max} - \alpha_t^{\min}} (s_2 - s_1) + s_1 \right) \tag{3}$$

where $\eta$ is the initial weight decay, and $(s_1, s_2)$ define the range of scaling ratios applied to $\eta$. $\alpha_t^i$ is the `PL_Alpha_Hill` value of module $i$ at step $t$, while $\alpha_t^{min}$ and $\alpha_t^{max}$ are the minimum and maximum `PL_Alpha_Hill` values among all modules at step $t$. Formula (3) guarantees that the adjusted weight decay, $f_t(i)$, remains within $[s_1\eta, s_2\eta]$ as a scaled variant of $\eta$.

We compare `AlphaDecay` with the `Uniform` baseline under varying weight decay settings, and present the results of perplexity and module-wise `PL_Alpha_Hill` values, as shown in figure 4. Notably, `AlphaDecay` assigns module-wise weight decay values in accordance with the imbalance observed in module-wise `PL_Alpha_Hill` metrics (see Figure 1), and reallocates these weight decays every 500 update steps. This dynamic assignment adaptively moderates the module-wise `PL_Alpha_Hill` imbalance present in the `Uniform` baseline by decreasing the elevated `PL_Alpha_Hill` values in MLP modules and increasing the lower values in attention-related modules (i.e., `att.v/o` and `att.q/k`). As a result, our method achieves consistently lower and more stable perplexity across different weight decay configurations, thereby improving model robustness and overall performance.

---

**Algorithm 1:** `AlphaDecay`

---

**Require :** initial weight decay $\eta$, number of training steps $T$, interval $\tilde{t}$ of using `AlphaDecay`,
        minimum and maximum scaling ratio $s_1, s_2$, and $\alpha_t^i$ refers to $i_{th}$ module's
        `PL_Alpha_Hill` at update step $t$
**for** $t \leftarrow 0$ **to** $T$ **do**
    **if** $mod(t, \tilde{t}) = 0$ **then**
        Compute $\alpha_t^i$ for all modules using the Hill estimator;
        Leverage all $\alpha_t^i$ and adopt $f_t(i) = \eta \cdot \left( \frac{\alpha_t^i - \alpha_t^{\min}}{\alpha_t^{\max} - \alpha_t^{\min}} (s_2 - s_1) + s_1 \right)$ to assign module-wise
        weight decay between $s_1\eta$ and $s_2\eta$;

---

## 4 Empirical results

In this section, we begin by presenting the complete experimental setup (Section 4.1), followed by a comparison between `AlphaDecay` and several baselines (Section 4.2). Finally, we analyze the impact

Table 1: Hyperparameters used in pre-training experiments.

| Model Size | LR | Tokens | Weight Decay | $(s_1, s_2)$ |
|---|---|---|---|---|
| 60M | 0.001 | 1B | 1e-5, 5e-6, 1e-6 | (0.67,3), (0.67,5), (0.67,5) |
| 135M | 0.001 | 2B | 1e-5, 5e-6, 1e-6 | (0.67,3), (0.67,5), (0.67,5) |
| 350M | 0.001 | 6B | 1e-5, 5e-6, 1e-6 | (0.67,3), (0.67,5), (0.67,5) |
| 1B | 0.0006 | 8.9B | 1e-5, 5e-6, 1e-6 | (0.67,3), (0.67,5), (0.67,5) |

Table 2: **(Main result).** Comparison with various weight decay scheduling strategies on pre-training various sizes of LLaMa models on C4 dataset. Validation perplexity ($\downarrow$) is reported. All baselines are carefully tuned. 'WD=0' indicates that weight decay is disabled during model training.

| Weight Decay | LLaMa-60M | | | LLaMa-135M | | | LLaMa-350M | | | LLaMa-1B | | |
|---|---|---|---|---|---|---|---|---|---|---|---|---|
| | 1e-5 | 5e-6 | 1e-6 | 1e-5 | 5e-6 | 1e-6 | 1e-5 | 5e-6 | 1e-6 | 1e-5 | 5e-6 | 1e-6 |
| WD=0 | | 33.23 | | | 24.60 | | | 18.62 | | | 16.11 | |
| Uniform | 32.39 | 32.56 | 33.03 | 22.99 | 23.14 | 24.14 | 17.12 | 16.74 | 17.50 | 15.36 | 14.66 | 15.03 |
| AWD[12] | 33.78 | 33.74 | 33.74 | 24.25 | 24.45 | 24.53 | 18.32 | 18.55 | 18.79 | 16.03 | 16.22 | 16.38 |
| Adadecay[30] | 32.24 | 32.52 | 33.03 | 23.20 | 23.08 | 23.96 | 18.21 | 17.42 | 17.91 | 17.23 | 18.14 | 15.35 |
| AlphaDecay | **31.56** | **31.58** | **32.61** | **22.76** | **22.55** | **23.49** | **17.00** | **16.66** | **16.88** | **15.13** | **14.55** | **14.63** |

of weight decay assignment functions, HT-SR module-wise metrics, PL fitting methods, and PL fitting time gaps through ablation studies (Section 4.4).

## 4.1 Experimental setup

**Models and Datasets**. We conduct a systematic evaluation of `AlphaDecay` across LLaMa-based architectures spanning four model scales (60M, 135M, 350M, and 1B parameters). All experiments employ the C4 dataset [31], a rigorously processed subset of Common Crawl widely adopted for language model pretraining. Our experimental design incorporates two key components: (1) a non-repeating data regime with sufficient tokens for convergence, and (2) standardized preprocessing pipelines across all model scales. This multi-scale approach facilitates systematic comparison of model behaviors across different capacity regimes, while minimizing potential confounding factors in the analysis.

**Hyperparameters.** The detailed hyperparameter settings for all model sizes are summarized in Table 1. All models are trained with Adam optimizer (gradient clipping at 1.0) and a cosine learning rate schedule, with 10% of the training tokens used for learning rate warmup. We conduct grid search over learning rates $\{0.01, 0.001, 0.0001\}$ and report the best configuration for each scale in the table. Weight decay settings and the corresponding $(s_1, s_2)$ parameter settings are also detailed in the table. `AlphaDecay` is performed every 500 update steps throughout all experiments.

## 4.2 LLM Pre-training

Table 2 presents the main results of our study, where we evaluate the effectiveness of different weight decay scheduling strategies on the pre-training of LLaMa models with varying parameter scales (60M, 135M, 350M, and 1B) on the C4 dataset. For each model size, we conduct comprehensive experiments across three commonly used weight decay values (1e-5, 5e-6, and 1e-6). Our proposed method is compared against several baselines, including the commonly used `Uniform` scheduling, adaptive global weight decay (`AWD`) [12], and adaptive per-module weight decay (`Adadecay`) [30]. All baseline methods are carefully tuned for a fair comparison.

**Observations.** ❶ **Weight Decay is Beneficial for Model Performance.** Comparing 'WD=0' (i.e., no weight decay) and `Uniform` across all model sizes, applying weight decay consistently leads to substantial reductions in validation perplexity. This provides empirical support for the importance and effectiveness of weight decay in LLM pre-training. ❷ **Superior and Consistent Gains Across All Weight Decay Settings.** `AlphaDecay` consistently yields the lowest validation perplexity across all evaluated weight decay settings (1e-5, 5e-6, 1e-6) and model sizes, surpassing both the `Uniform` baseline and the adaptive weight decay methods (`AWD` and `Adadecay`). This consistent superiority across various regularization strengths demonstrates the robustness of our approach and underscores its potential applicability in LLM pre-training. ❸ **Scalability to Larger Models**. The performance

improvements achieved by `AlphaDecay` are consistently observed from the smallest (60M) to the largest (1B) parameters, indicating the scalability and generality of our approach.

Furthermore, our experiments reveal that existing adaptive weight decay methods, originally designed for architectures without attention components, such as `AWD` and `Adadecay`, do not yield optimal results for LLMs. This may be attributed to their lack of consideration for the distinct characteristics and optimization requirements of attention and MLP modules within transformer architectures. In contrast, our approach is, to the best of our knowledge, the first to demonstrate that a tailored weight decay scheduling strategy can consistently enhance LLM training by explicitly accounting for the heterogeneous characteristics of different modules.

### 4.3 Downstream tasks & architectures

This section introduces our evaluation of downstream gains across zero-shot commonsense reasoning and fine-tuning tasks, with results on additional model architecture and image classification task (GPT-nano/C4 perplexity; ViT-tiny/ImageNet-1K Top-1).

**Zero-shot Results.** We evaluate the pretrained LLaMa-1B checkpoints from Table 2 on seven zero-shot commonsense reasoning tasks using lm-eval-harness with its default prompts. As shown in Table 3, `AlphaDecay` delivers the best results on 6 of 7 benchmarks, indicating that its pretraining gains transfer well to downstream reasoning tasks and support broad applicability.

Table 3: **(Zero-shot results of commonsense-reasoning tasks).** Zero-shot evaluation results (↑) on seven commonsense reasoning benchmarks using the LLaMa-1B model pretrained with different methods.

| Method | ARC-c | ARC-e | PIQA | Hellaswag | OBQA | Winogrande | BOOLQ | Avg. |
|--------|-------|-------|------|-----------|------|------------|-------|------|
| Uniform | 20.22 | 46.72 | 67.68 | 32.94 | 18.8 | 49.41 | 54.74 | 41.50 |
| AdaDecay | 19.20 | 46.72 | 66.97 | 32.96 | 18.0 | **51.54** | 56.36 | 41.68 |
| AWD | 19.18 | 46.34 | 66.65 | 31.37 | 18.0 | 51.07 | 57.25 | 41.41 |
| AlphaDecay | **20.90** | **48.86** | **68.44** | **34.16** | **19.80** | 50.59 | **60.70** | **43.35** |

**Finetuning Results.** We evaluate all baselines on GLUE finetuning tasks with roberta-base. As reported in Table 4, `AlphaDecay` attains the top result on 6 of 7 tasks. This evidences that `AlphaDecay` is effective not only during pretraining but also transfers well to finetuning settings.

Table 4: **(Finetuning tasks).** Finetuning results (↑) on eight benchmarks from the GLUE dataset using `roberta-base` with different methods.

| Method | cola | mnli | mrpc | qnli | qqp | rte | sst2 | stsb | Avg. |
|--------|------|------|------|------|-----|-----|------|------|------|
| Uniform | 59.73 | 86.78 | 87.01 | 92.59 | 89.97 | 70.11 | 93.69 | 90.78 | 83.83 |
| AdaDecay | 60.45 | 87.23 | 88.19 | 92.62 | 89.95 | 73.36 | 93.73 | 90.9 | 84.55 |
| AWD | 60.72 | **87.44** | 89.53 | 92.58 | 90.08 | 72.27 | 93.72 | 90.9 | 84.66 |
| AlphaDecay | **62.82** | 87.11 | **89.61** | **92.73** | **90.12** | **73.86** | **93.77** | **90.91** | **85.12** |

**Across architectures and datasets.** We evaluate all methods on GPT-nano/C4 (perplexity) and ViT-tiny/ImageNet-1K (Top-1 accuracy). As shown in Table 5, our method attains the lowest perplexity on GPT-nano/C4 and the highest Top-1 accuracy on ViT-tiny/ImageNet-1K, outperforming `Uniform`, `AWD`, and `AdaDecay`. These results demonstrate consistent effectiveness across different architectures and datasets, and strong generalization beyond a single setting.

Table 5: **(Across architectures and datasets).** Results on GPT-nano/C4 (Perplexity) and ViT-tiny/ImageNet-1K (Top-1) with different methods.

| Backbone / Dataset | Metric | Uniform | AWD | AdaDecay | AlphaDecay |
|--------------------|--------|---------|-----|----------|------------|
| **GPT-nano / C4** | PPL(↓) | 27.56 | 27.64 | 27.68 | **27.37** |
| **ViT-tiny / ImageNet-1K** | Top-1(↑) | 66.41% | 64.98% | 66.26% | **67.73%** |

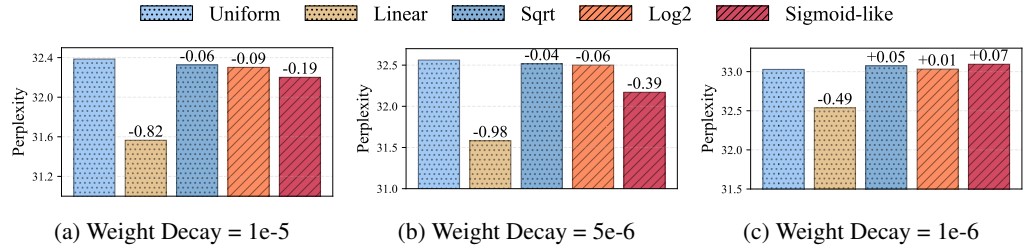

Figure 5: **(Varying weight decay assignment functions).** Results of using different weight decay assignment functions under different weight decay settings. All experiments are conducted on LLaMa-60M. The value on the top of each bar indicates the difference from the leftmost bar in each plot and the same processing is applied in Figure 6, Figures 7, and Figures 8.

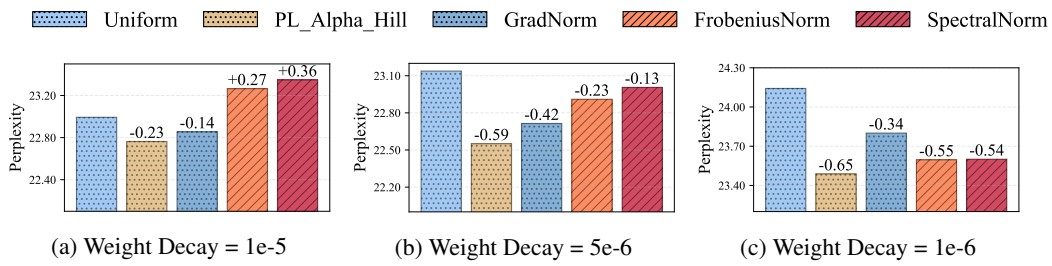

Figure 6: **(Varying HT-SR metrics).** Comparing `PL_Alpha_Hill` with multiple HT-SR metrics under different weight decay settings. All experiments are conducted on LLaMa-135M.

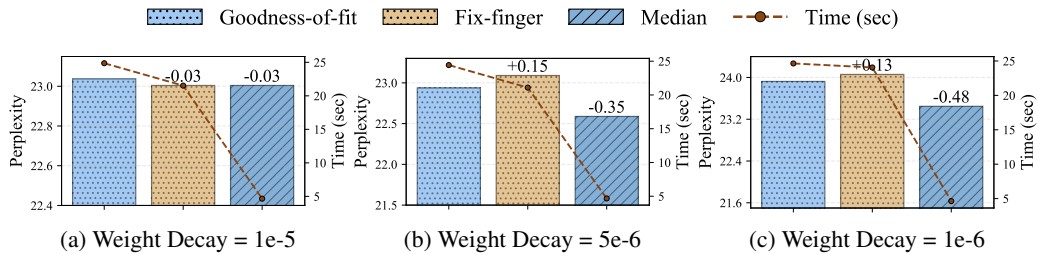

Figure 7: **(Varying PL fitting methods).** Comparison of various PL fitting methods. The bar plot and left y-axis represent perplexity (lower the better), while the line plot and right y-axis indicate the time taken for `AlphaDecay` once (in seconds, lower the better). The computation times are averaged over all PL fitting operations throughout the model training process. All experiments are conducted using LLaMa-135M.

## 4.4 Analysis

**Varying Weight Decay assignment functions.** We examine the performance of `PL_Alpha_Hill` with different weight decay assignment functions, which determine the allocation ratios of weight decay across different modules. Figure 5 presents the results obtained by different assignment functions: `Uniform`, `Linear`, `Sqrt`, `Log2`, and `Sigmoid-like`. Among these, `Linear` achieves the best results across all weight decay settings, showing a notable advantage over other methods.

**Varying HT-SR metrics.** To investigate the impact of various HT-SR metrics on regulating weight decay during model training, we conducted ablation studies comparing these metrics. While prior work has primarily utilized `GradNorm` [30; 20; 42] and `FrobeniusNorm` [12; 16] as indicators for adjusting weight decay, our study further evaluates additional metrics, including `PL_Alpha_Hill` and `SpectralNorm`, under the same experimental settings. Results in Figure 6 show that most HT-SR metrics outperform the `uniform` baseline, while `PL_Alpha_Hill` achieves the lowest perplexity (lower the better) among all evaluated methods.

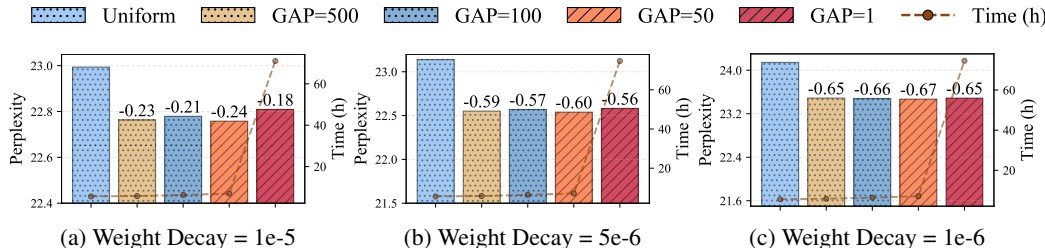

Figure 8: **(Varying PL fitting gaps).** We conduct PL fitting at varying specified gaps of training steps. The bar plot and left y-axis represent perplexity (lower the better), while the line plot and right y-axis indicate the time required for training completion (lower the better). The computation times reflect the NVIDIA A100 hours utilized for completing model training. All experiments are conducted using LLaMa-135M.

**Varying PL fitting methods.** In our proposed framework, the HT-SR metric `PL_Alpha_Hill` is employed to guide the projection-based adjustment of weight decay during training. Since `PL_Alpha_Hill` is derived through PL fitting, and the choice of fitting method can influence both computational efficiency and the final training effectiveness, we conduct an ablation study to systematically assess its impact. Figure 7 presents a comparative analysis of three PL fitting methods— `Goodness-of-fit` [1; 29; 6], `Fix-finger`[43], and `Median`[46] —across multiple weight decay values. Across all settings, `Median` not only ensures optimal training performance but also notably decreases computation time compared to the other approaches, making it the preferred choice for PL fitting within our method.

**Varying PL fitting gaps.** To further analyze the stability of our proposed approach, we investigate the impact of varying update gaps for weight decay adjustments during training. It is computationally inefficient to update weight decay at every training step. Thus, we explore the performance of our method by updating weight decay at different training step gaps: 1, 50, 100, and 500 steps. Figure 8 shows that our approach achieves stability across all gap settings. Notably, across all weight decay settings, using training step intervals from 1 to 500 consistently outperforms the `Uniform` setting, including when the interval is as large as 500 training steps. This demonstrates the robustness of our method to update frequency. Therefore, we select a gap of 500 in all experiments because it provides substantial computational savings while maintaining stable and competitive model performance across various settings.

## 5 Conclusion

Weight decay is a standard regularization technique in deep learning, typically implemented with a single decay rate for all parameters. However, this uniform application lacks theoretical justification and may not be optimal. We present a systematic study of module-wise weight decay scheduling, an overlooked but important aspect of model regularization. The proposed `AlphaDecay` framework provides a principled approach to module-specific decay rates based on HT-SR theory. Through extensive experiments, we demonstrate that `AlphaDecay` consistently improves model performance across different pretraining scales. To our knowledge, this is the first work to formally investigate and establish a framework for module-level weight decay scheduling in LLMs. Our results indicate that weight decay scheduling represents a promising direction for future research. While this work represents an initial exploration, it opens new possibilities for understanding and improving weight decay in LLMs.

**Limitations.** While our study offers initial insights into module-wise weight decay scheduling, several limitations remain. First, evaluation on Mixture-of-Experts (MoE) models is left for future work. Second, interactions with other regularization and optimization techniques are yet to be systematically assessed. Addressing these issues represents valuable directions for future research.

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

# Appendix

## A  Details of Experiments

### A.1  Architecture

To ensure reproducibility and consistency with prior research ([45; 22]), we adopt the LLaMa architectural specifications and pre-training hyperparameters as detailed in Table 6. All model variants are trained with a uniform maximum sequence length of 256, a batch size of 512, and an aggregate of 13K tokens per batch.

Table 6: Hyperparameters of LLaMa models used in this paper.

| Params | Hidden | Intermediate | Heads | Layers | Steps | Data amount | LR | Batch Size |
|---|---|---|---|---|---|---|---|---|
| 60M | 512 | 1376 | 8 | 8 | 10K | 1B | $1 \times 10^{-3}$ | 512 |
| 135M | 768 | 2048 | 12 | 12 | 20K | 2B | $1 \times 10^{-3}$ | 512 |
| 350M | 1024 | 2736 | 16 | 24 | 60K | 6B | $1 \times 10^{-3}$ | 512 |
| 1B | 2048 | 5461 | 32 | 24 | 90K | 9B | $6 \times 10^{-4}$ | 512 |

### A.2  Hyperparameter settings for reproducing our figures

We report all hyperparameters and all numerical values of experimental results shown in the main paper. First, Table 7 reports the details of experiments shown in Figure 3 and Figure 4. Then, Table 8, Table 9, Table 10 and Table 11 respectively report the details of the experiments shown in Figure 5, Figure 6, Figure 7 and Figure 8.

Table 7: Parameter settings of the experiment reported in Section 3.3 Figure 3 and Figure 4.

| Method | Model Size | Weight Decay | Perplexity | Scaling Ratio $(s_1, s_2)$ | Method | Model Size | Weight Decay | Perplexity | Scaling Ratio $(s_1, s_2)$ |
|---|---|---|---|---|---|---|---|---|---|
| Uniform | LLaMa 60M | 3e-6 | 33.306 | - | Uniform | LLaMa 135M | 3e-6 | 23.604 | - |
| | | 4e-6 | 33.219 | - | | | 4e-6 | 23.437 | - |
| | | 5e-6 | 33.131 | - | | | 5e-6 | 23.272 | - |
| | | 6e-6 | 33.157 | - | | | 6e-6 | 23.161 | - |
| | | 7e-6 | 33.077 | - | | | 7e-6 | 23.136 | - |
| | | 8e-6 | 33.028 | - | | | 8e-6 | 23.109 | - |
| | | 9e-6 | 33.008 | - | | | 9e-6 | 23.083 | - |
| | | 1e-5 | 33.002 | - | | | 1e-5 | 23.111 | - |
| AlphaDecay | LLaMa 60M | 3e-6 | 32.557 | (0.67,5) | AlphaDecay | LLaMa 135M | 3e-6 | 22.895 | (0.67,5) |
| | | 4e-6 | 32.364 | (0.67,5) | | | 4e-6 | 22.812 | (0.67,5) |
| | | 5e-6 | 32.171 | (0.67,5) | | | 5e-6 | 22.730 | (0.67,5) |
| | | 6e-6 | 32.144 | (0.67,5) | | | 6e-6 | 22.770 | (0.67,5) |
| | | 7e-6 | 32.122 | (0.67,5) | | | 7e-6 | 22.772 | (0.67,3) |
| | | 8e-6 | 32.077 | (0.67,5) | | | 8e-6 | 22.830 | (0.67,3) |
| | | 9e-6 | 32.097 | (0.67,5) | | | 9e-6 | 22.885 | (0.67,3) |
| | | 1e-5 | 32.099 | (0.67,5) | | | 1e-5 | 22.934 | (0.67,3) |

Table 8: Parameter settings of the experiment reported in Section 4.4 Figure 5. All experiments are conducted on LLaMa-60M.

| Weight Decay | Uniform | Linear | Sqrt | Log2 | Sigmoid-like | Scaling Ratio $(s_1, s_2)$ |
|---|---|---|---|---|---|---|
| 1e-5 | 32.386 | **31.565** | 32.326 | 32.301 | 32.201 | (0.67,3) |
| 5e-6 | 32.562 | **31.582** | 32.517 | 32.501 | 32.171 | (0.67,5) |
| 1e-6 | 33.028 | **32.537** | 33.074 | 33.033 | 33.095 | (0.67,5) |

Table 9: Parameter settings of the experiment reported in Section 4.4 Figure 6. All experiments are conducted on LLaMa-135M.

| Weight Decay | Uniform | PL_Alpha_Hill | GradNorm | FrobeniusNorm | SpectralNorm | Scaling Ratio $(s_1, s_2)$ |
|---|---|---|---|---|---|---|
| 1e-5 | 22.993 | **22.763** | 22.855 | 23.265 | 23.348 | (0.67,3) |
| 5e-6 | 23.138 | **22.551** | 22.714 | 22.91 | 23.006 | (0.67,5) |
| 1e-6 | 24.142 | **23.488** | 23.801 | 23.596 | 23.601 | (0.67,5) |

Table 10: Parameter settings of the experiment reported in Section 4.4 Figure 7. The computation times are averaged over all PL fitting operations throughout the model training process.

| Model Size | Weight Decay | Goodness-of-fit | | Fix-finger | | Median | | Scaling Ratio $(s_1, s_2)$ |
|---|---|---|---|---|---|---|---|---|
| | | Perplexity | Computation Time (sec) | Perplexity | Computation Time (sec) | Perplexity | Computation Time (sec) | |
| LLaMa -60M | 1e-5 | **32.166** | 8.73±0.30 | 32.231 | 7.90±0.33 | **31.628** | 1.67±0.01 | (0.67,3) |
| | 5e-6 | **32.436** | 10.62±0.04 | 32.381 | 9.22±0.47 | **31.614** | 1.69±0.01 | (0.67,5) |
| | 1e-6 | 32.993 | 8.28±0.03 | 33.059 | 7.80±0.67 | **32.703** | 1.66±0.01 | (0.67,5) |
| LLaMa -135M | 1e-5 | **22.937** | 24.86±0.11 | 23.004 | 21.53±0.83 | 23.004 | 4.67±0.02 | (0.67,3) |
| | 5e-6 | 22.937 | 24.43±0.11 | 23.090 | 22.00±0.86 | **22.588** | 4.68±0.03 | (0.67,5) |
| | 1e-6 | 23.924 | 24.64±0.13 | 24.058 | 21.90±0.80 | **23.448** | 4.63±0.01 | (0.67,5) |

Table 11: Parameter settings of the experiment reported in Section 4.4 Figure 8. The computation times reflect the NVIDIA A100 hours utilized for completing model training.

| Model Size | Weight Decay | Uniform | GAP=500 | GAP=250 | GAP=100 | GAP=50 | GAP=1 | Scaling Ratio $(s_1, s_2)$ |
|---|---|---|---|---|---|---|---|---|
| LLaMa -60M | 1e-5 | 32.386 | 31.614 | 31.628 | **31.555** | 31.618 | 31.594 | (0.67,3) |
| | 5e-6 | 32.562 | **31.628** | 31.633 | 31.673 | 31.717 | 31.712 | (0.67,5) |
| | 1e-6 | 33.029 | 32.703 | 32.718 | 32.754 | **32.663** | 32.769 | (0.67,5) |
| | Computation Time | 1.4h | 1.4h | 1.4h | 1.5h | 1.6h | 9.3h | |
| LLaMa -135M | 1e-5 | 22.994 | 22.763 | **22.756** | 22.779 | 22.758 | 22.809 | (0.67,3) |
| | 5e-6 | 23.138 | 22.551 | **22.537** | 22.569 | 22.539 | 22.581 | (0.67,5) |
| | 1e-6 | 24.142 | 23.488 | 23.477 | 23.479 | **23.468** | 23.488 | (0.67,5) |
| | Computation Time | 5.6h | 5.7h | 5.9h | 6.3h | 7.1h | 74.5h | |

## A.3 Assignment Function Formulas

For `AlphaDecay`, we selected the linear interpolation (fomula 3) for weight decay assignment function $f_t$, based on its superior performance in our ablation study. We provide the remaining assignment functions here:

- `Sqrt` : $f_t(i) = \eta \frac{\sqrt{\alpha_t^i}}{\frac{1}{L}\sum_{j=1}^{L}\sqrt{\alpha_t^j}}$

- `Log2` : $f_t(i) = \eta \frac{log2(\alpha_t^i)}{\frac{1}{L}\sum_{j=1}^{L} log2(\alpha_t^j)}$

- `sigmoid-like` : $f_t(i) = \eta \frac{2}{1+\exp(-\beta \tilde{g}_j^t)}$ with $\tilde{g}_j^t = (|g_j^t| - \mu_l^t)/\sigma_l^t$

Here, $\eta$ denotes the initial weight decay, $\alpha_t^i$ is `PL_Alpha_Hill` of the module $i$ at step $t$, and $L$ is the total number of model modules. $\mu_l^t$ is the mean of gradient norms $|g_j^t|$ in the layer $l$; $\sigma_l^t$ is the std of gradient norms $|g_j^t|$ in the layer $l$; $\beta = 4$ is a control parameter for the steepness of function value transition.

## A.4 Derivation of Formula 2

We provide the derivation for estimating the power-law exponent $\alpha$ from empirical singular value data, which is central to our approach. We assume the empirical distribution follows a power-law (Pareto) distribution:

$$p(x) = cx^{-\alpha}$$

For normalization over $x \geq x_{min}$:

$$\int_{x_{\min}}^{\infty} p(x)\,dx = 1 \implies c = (\alpha - 1)x_{\min}^{\alpha-1}$$

Thus, the probability density function (PDF) becomes:

$$p(x) = (\alpha - 1)x_{\min}^{\alpha-1} x^{-\alpha} = \frac{\alpha - 1}{x_{\min}} \left( \frac{x}{x_{\min}} \right)^{-\alpha}$$

Given a set of observed data $x_1, x_2, \ldots, x_n$ with $x_i \geq x_{min}$, the likelihood function is:

$$p(x_1, x_2, \ldots, x_n) = \prod_{i=1}^{n} p(x_i) = \prod_{i=1}^{n} \frac{\alpha - 1}{x_{\min}} \left( \frac{x_i}{x_{\min}} \right)^{-\alpha}$$

The log-likelihood is therefore:

$$\mathcal{L} = \sum_{i=1}^{n} \left[ \ln(\alpha - 1) - \ln x_{\min} - \alpha \ln \frac{x_i}{x_{\min}} \right]$$

To obtain the maximum likelihood estimate, we set the derivative of $\mathcal{L}$ with respect to $\alpha$ to zero:

$$\frac{\partial \mathcal{L}}{\partial \alpha} = 0 \implies \alpha = 1 + n \left[ \sum_{i=1}^{n} \ln \frac{x_i}{x_{\min}} \right]^{-1}$$

This result is known as the standard **Hill estimator**, and we denote the fitted exponent as `PL_Alpha_Hill`.

## B More experiments

### B.1 LLM Pre-training with AdamW

Table 12 provides a comparison of several weight decay scheduling strategies for pre-training LLaMa-60M and LLaMa-130M models with the AdamW optimizer. The results clearly demonstrate the effectiveness of applying weight decay, as all scheduling strategies outperform the baseline with no weight decay (WD=0) in terms of validation perplexity.

Table 12: **(AdamW.)** Comparison of various weight decay scheduling strategies using AdamW optimizer for pre-training LLaMa-60M and LLaMa-130M models under different weight decay values. Validation perplexity ($\downarrow$) on the C4 dataset is reported. All baselines are carefully tuned. 'WD=0' indicates that weight decay is disabled during model training.

| Weight Decay | LLaMa-60M | | | LLaMa-135M | | |
|---|---|---|---|---|---|---|
| | 0.1 | 0.05 | 0.01 | 0.1 | 0.05 | 0.01 |
| WD=0 | | 32.73 | | | 24.39 | |
| Uniform | 31.95 | 32.31 | 32.66 | 23.32 | 23.81 | 24.28 |
| AWD | 32.58 | 32.67 | 32.67 | 24.30 | 24.35 | 24.41 |
| Adadecay | 31.88 | 32.25 | 32.58 | 23.18 | 23.62 | 24.21 |
| AlphaDecay | 31.20 | 31.65 | 32.45 | 22.66 | 23.04 | 23.98 |

`AlphaDecay` consistently outperforms other weight decay scheduling strategies across different model sizes and hyperparameter settings, demonstrating superior regularization and generalization when training with AdamW. These results highlight the robustness and effectiveness of `AlphaDecay`, supporting its adoption for optimizing large-scale transformer-based language models.

### B.2 Dependent t-test for paired samples

Table 13 provides a comparison of several weight decay scheduling strategies using the Adam optimizer, evaluated through repeated experiments with different random seeds.

Table 13: **(Dependent t-test with Adam.)** Each method (`Uniform`, `AWD`, `AdaDecay`, and `AlphaDecay`) is evaluated by conducting six repeated experiments with random seeds { 5, 6, 7, 8, 9, 10 }. Validation perplexity is reported as mean ± standard deviation. For each weight decay setting, a dependent t-test for paired samples is performed, comparing `AlphaDecay` against `Uniform`, `AWD`, and `AdaDecay`, respectively. The resulting p-values are presented alongside perplexity scores.

| Method | Weight Decay=0 Perplexity | Weight Decay=1e-5 Perplexity | P-value | Weight Decay=5e-6 Perplexity | P-value | Weight Decay=1e-6 Perplexity | P-value |
|---|---|---|---|---|---|---|---|
| `Uniform` | | $22.97 \pm 0.07$ | 8.38e-4 | $23.12 \pm 0.03$ | 1.47e-6 | $24.12 \pm 0.04$ | 1.05e-7 |
| `AWD` | $24.55 \pm 0.07$ | $24.13 \pm 0.15$ | 6.94e-6 | $24.46 \pm 0.07$ | 5.34e-8 | $24.53 \pm 0.03$ | 5.34e-9 |
| `AdaDecay` | | $23.18 \pm 0.05$ | 1.46e-5 | $23.07 \pm 0.04$ | 1.11e-7 | $24.00 \pm 0.03$ | 2.69e-9 |
| `AlphaDecay` | | $22.77 \pm 0.02$ | | $22.54 \pm 0.03$ | | $23.44 \pm 0.02$ | |

The results demonstrate the benefit of applying weight decay for improved validation perplexity, with `AlphaDecay` consistently exhibiting superior performance and stability across all tested settings. The dependent t-test results further substantiate these findings, with statistically significant p-values supporting the advantage of `AlphaDecay` over `Uniform`, `AWD`, and `AdaDecay` in nearly all cases.

## B.3 Varying assignment function hyperparameters

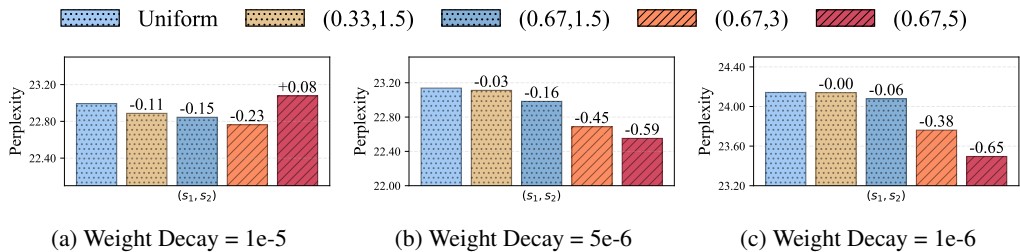

(a) Weight Decay = 1e-5  (b) Weight Decay = 5e-6  (c) Weight Decay = 1e-6

Figure 9: **(Hyperparameter study on $(s_1, s_2)$).** Search for hyperparameters $(s_1, s_2)$ of different weight decay settings on C4. The hyperparameter choice used in the paper (see table 1) performs best among all the cases. All experiments are conducted on LLaMa-135M. The value on the top of each bar indicates the difference from the leftmost bar in each plot.

We provide additional results of a hyperparameter study on $(s_1, s_2)$, in which we consider four different settings for $(s_1, s_2)$: [(0.33, 1.5), (0.67, 1.5), (0.67, 3), (0.67, 5)]. We run tasks on C4 with LLaMa-135M. Our results in figure 9 show that using a larger weight decay scaling range, such as (0.67, 3) or (0.67, 5), yields the best performance. This hyperparameter setting is the default setting used in our paper. All hyperparameters are consistent with those described in the main paper.

