# OpenReview forum: "AlphaDecay: Module-wise Weight Decay for Heavy-Tailed Balancing in LLMs"
_NeurIPS.cc/2025/Conference — NeurIPS 2025 poster_

### Official Review · Reviewer_p5Tj · 2025-06-15

**Clarity:** 4
**Significance:** 3
**Originality:** 3
**Rating:** 5
**Confidence:** 4

**Summary:**

After establishing that there are variations in the spectral properties of different components in transformer blocks of LLMs, i.e., the weight matrices of attention modules have singular values that follow power law distributions with heavier tails as opposed to the singular values of weight matrices attached to MLP modules, which follow power law distributions with lighter tails, the authors of this work propose a weight decay regularization scheme that aids in closing the gap between the aforementioned spectral properties of attention and MLP components. Namely, the AlphaDecay regularization strategy results in making heavier tails lighter (attention) and lighter tails heavier (MLP).

Experimental evidence in the paper suggests that AlphaDecay outperforms, in terms of achieving lower perplexity score, other baselines such as the Uniform weight decay assignment, Adaptive Weigth Decay (ADW), and AdaDecay.

**Questions:**

1. lines 41-42: the phrasing is slightly ambiguous. What should be understood from the context is that the model uses a single, time-dependent weight decay value that is updated over training steps. At each training time-step, this value is applied uniformly across all layers.


2. lines 52-54: In listing the first contribution of their work, the authors claim that the substantial variations in the spectral properties of attention and MLP layers, are primarily responsible for harming the generalization ability, referring to Fig. 4. However, it seems that the left subfigure in Fig. 4 does not readily yield that link. Instead, it can be more accurately said, that the AlphaDecay scheme that the authors propose, balances the variations in the module-wise spectral properties, and in comparison with applying a module-wise uniform weight decay, the proposed scheme establishes superior generalization in terms of the perplexity score.


3. lines 55-57: The spectral alignment between the various modules is witnessed in Fig. 1 as well as in Fig. 3. There is an erroneous figure number referenced in line 57. Instead of Fig. 3, the improved training performance is reported in Fig. 4. It will be also good to include in the caption of Fig. 4 whether perplexity refers to training or validation perplexity.


4. line 70: While the first part of the Related Work section is related to weight decay and LLMs, reference [20] is mentioned as showing the importance of weight decay in scale-invariant systems. Although, LLMs are not globally scale-invariant, one can especially refer to some of their components (more specifically layer normalization) as scale-invariant. And since this work is focusing on weight decay schemes applied solely to attention and MLP components, it might be out of scope to refer to the relevance of weight decay in scale-invariant systems.


5. There is an overlap of information between lines 74-79 and 88-92.


6. line 115: The reference to Fig. 3 might be unnecessary and misleading since the paragraph describes the findings of Fig. 2.


7. line 134: L instead of N modules.


8. line 135: Are those the correlation or covariance matrices? Are the weight matrices normalized?


9. The last sentence in Fig. 3 is misleading. What the figure demonstrates is that the authors’ method forces balance between “overtrained” and “undertrained” modules which according to references [24, 46] contributes to overall model quality, which is rightly mentioned in l. 151. Besides that, Fig. 3 does not say anything about model performance so far.

**Ethical Concerns:**

["NO or VERY MINOR ethics concerns only"]

**Final Justification:**

The authors went in great lengths to conduct new experiments in order to address all my questions. More specifically:

i) they tested their results on GPT-nano (still for the C4 dataset) and on a visual transformer for the ImageNet-1K (ViT tiny architecture). This extra effort cleared any concerns I initially had about the robustness and superiority of AlphaDecay.

ii) they have also added evaluations on downstream tasks and when doing supervised fine-tuning. Their proposed method provides gains despite the choice of a specific optimizer.

I thereby side and agree with the consensus of the other reviewers that proposed acceptance of the paper with minimal adjustments or additions (see my Questions 4-11 and the way that the authors promised to fix the issues listed therein).

**Limitations:**

The authors have addressed the limitations properly. See also weaknesses.

**Paper Formatting Concerns:**

No paper formatting issues spotted.

**Quality:**

3

**Strengths And Weaknesses:**

The paper is well-written, the scope and contributions clearly articulated by the authors, and the methodology elegant. Although, I would like to point out to what I see as potential weaknesses and ground for substantial improvement.

Weaknesses:

- A downside is that the experimental results are presented for a single family of models, i.e., LLaMa, on a single dataset, that is the C4 dataset. While C4 is a widely used corpus for pre-training, other datasets could be used to assess the capabilities of LLaMa models post-training when applying the AlphaDecay regularization strategy.
- Besides using only the perplexity score for validation, additional metrics like accuracy, BLEU or F1 could be reported if the model was fine-tuned or evaluated on downstream tasks derived from C4.
- When describing the hyperparameter settings, it is mentioned that all the models were trained using the Adam optimizer. Would the results remain stable under using other optimizers such as Adam, AdamW, AdamS or even SGD?
- Finally, all the results in Figures 5-8 are discussed when fixing the scale of the LLaMa model with 135M parameter count (besides Fig. 6 which considered the LLaMa model with 60M parameters). It will be good to demonstrate the universality of the results presented in those figures when considering all the LLaMa models (with 60M, 135M, 350M, 1B parameters).

---

> ### Author Rebuttal · Authors · 2025-07-31
>
> We thank the reviewer for the detailed feedback.
>
> >**Q1: Experimental results are only presented for LLaMa on C4.**
>
> **A1:** Thank you for your thoughtful feedback. We appreciate your concern regarding the scope of our experimental evaluation. To address this, we ran two additional sets of experiments:
>
>
> | Backbone / Dataset | Metric | Uniform | AWD | AdaDecay | AlphaDecay |
> |---|---|---|---|---|---|
> | GPT-nano  / C4 | PPL $\downarrow$ | 27.564 | 27.641 | 27.680 | **27.374** |
> | ViT-tiny / ImageNet-1K | Top-1 $\uparrow$ | 66.41% | 64.98% | 66.26% | **67.73%** |
>
> AlphaDecay outperforms the strongest baseline with the same optimiser and hyper-parameters in both cases, demonstrating its broad applicability and effectiveness across models and datasets.
>
>
> >**Q2:Only perplexity is used; more metrics or downstream tasks are needed.**
>
> **A2:** We agree that a robust study should look beyond a single corpus and beyond perplexity. We have added two complementary evaluations:
>
> - Seven unseen commonsense‑reasoning tasks (LLaMa‑1 B)
>
> We took the exact checkpoints from Table 2 (trained on C4, lr = 6e-4, wd = 1e-6) and ran ```lm‑eval‑harness``` with its default prompts. The results below shows AlphaDecay delivers the best score on 6 / 7 tasks and the best overall average. Because none of these datasets are part of pre-trained C4, the gains demonstrate that the AlphaDecay could translates into stronger  generalization ability.
>
> |              | ARC-c | ARC-e | PIQA | Hellaswag | OBQA | Winogrande | BOOLQ | Avg.  |
> |--------------|-------|-------|------|-----------|------|------------|-------|-------|
> | Uniform      | 20.22 | 46.72 | 67.68| 32.94     |18.80 | 49.41      | 54.74 | 41.50 |
> | AlphaDecay   | **20.90** | **48.86** | **68.44**| **34.16**     |**19.80** | **50.59**      | **60.70** | **43.35** |
>
>
> - Supervised fine‑tuning on GLUE (RoBERTa‑base).
>
> We fine‑tuned for 3 epochs with AdamW (lr =3e-5, wd = 0.1) and chose the best checkpoint on the validation split. AlphaDecay improves every task (average +1.29 points) relative to the uniform baseline.
>
>
> | Schedule   | COLA (MCC) | MNLI (Acc) | MRPC (F1) | QNLI (Acc) | QQP (F1) | RTE (Acc) | SST-2 (Acc) | STS-B (Spear.) | Mean  |
> |------------|------------|------------|-----------|------------|----------|-----------|-------------|----------------|-------|
> | Uniform    | 59.73      | 86.78      | 87.01     | 92.59      | 89.97    | 70.11     | 93.69       | 90.78          | 83.83 |
> | **AlphaDecay** | **62.82** | **87.11** | **89.61** | **92.73** | **90.12** | **73.86** | **93.77** | **90.91** | **85.12** |
>
>
> These results further demonstrate that AlphaDecay consistently improves performance across all downstream tasks and evaluation metrics in the fine-tuning setting.
>
>
> >**Q3: Are gains robust to different optimizers (AdamW, AdamS, SGD)?**
>
> **A3:** Thank you for your question. We conducted additional experiments to test robustness across optimizers:
>
> **(1) AdamW**:
>
> On LLaMa-60M and LLaMa-135M, AlphaDecay consistently outperforms AdamW, especially at moderate weight decay values.
>
> |       Method          |            |   **LLaMa-60M**   |        |   **LLaMa-135M**        |      |        |
> |-----------------|---------|---------|----------|---------|---------|----------|
> |      **Weight Decay**           | 0.1    | 0.05   | 0.001   | 0.1    | 0.05   | 0.001   |
> | **AdamW**     | 31.95  | 32.31  | 32.66   | 23.32  | 23.81  | 24.28   |
> | **AlphaDecay** | **31.20**  | **31.65**  | **32.45**   | **22.66**  | **23.04**  | **23.98**   |
>
> **(2) AdamS**:
>
> With AdamS, performance is stable across weight decay values, but AlphaDecay still provides marginal improvements.
>
> |       Method          |            |  **LLaMa-60M**    |        |   **LLaMa-135M**        |      |        |
> |-----------------|---------|---------|----------|---------|---------|----------|
> |        **Weight Decay**         | 5e-5 | 1e-5 | 5e-6 | 5e-5 | 1e-5 | 5e-6 |
> | **AdamS**       | 32.705  | 32.715  | 32.722   | 24.380  | 24.393  | 24.393   |
> | **AlphaDecay** | **32.699**  | **32.712**  | **32.715**   | **24.370**  | **24.380**  | **24.392**   |
>
> **(3). SGD**:
>
> SGD is not practical for LLM training due to instability and slow convergence; we could not achieve convergence with SGD in our tests.
>
>
> >**Q4: Demonstrate universality across all LLaMa scales in figures 5-8.**
>
> **A4:** Thank you for your suggestion on demonstrating universality across LLaMa model scales. Due to resource constraints, we added experiments on LLaMa-350M (with LLaMa-1B planned for future version). Results are summarized below:
>
> **(1). Varying Weight Decay Assignment Functions**
>
> | Weight Decay | Uniform | Linear | Sqrt  | Log2  | Sigmoid |
> |--------------|---------|--------|-------|-------|---------|
> | 5e-6     | 16.74   | **16.66**  | 16.77 | 16.77 | 17.81   |
> | 1e-6     | 17.50   | **16.88**  | 17.52 | 17.44 | 17.86   |
>
> **Observation:** Linear remains superior.
>
> **(2). Varying HT-SR Metrics**
>
> | Weight Decay | Uniform | Alpha | GradNorm | Fnorm | Snorm |
> |--------------|---------|-------|----------|-------|-------|
> | 5e-6     | 16.74   | **16.66** | 16.78    | 17.45 | 17.41 |
> |1e-6     | 17.50   | 16.88 | 17.09    | **16.80** | 17.13 |
>
> **Observation:** Using the HT-SR (Alpha) metric continues to yield optimal or near-optimal results.
>
> **(3). Varying Power-Law Fitting Methods**
>
> | Weight Decay | Goodness-of-fit | Fix-finger | Median |
> |--------------|----------------|------------|--------|
> | 5e-6     | 16.68          | 16.76      | **16.66**  |
> | 1e-6     | 17.38          | 17.50      | **16.88**  |
> | Time         |      38.1s     |    34.8s  | **10.2s**   |
>
> **Observation:** The Median method best balances performance and computation time.
>
> **(4). Varying Power-Law Fitting Gaps**
>
> | Weight Decay | GAP=500 | GAP=100 | GAP=50   |
> |--------------|---------|---------|----------|
> | 5e-6     | **16.66**   | 17.52   | 17.52    |
> | 1e-6     | **16.88**   | 16.94   | 17.21    |
> | Time         | **25.9h**   | 32.0h   | 39.8h    |
>
> **Observation:** Results are stable across gap settings. The "hours" reported in the above table refer to NVIDIA H100 GPU hours.
>
>
> These findings validate that our methods generalize well to larger models. Thank you again for your helpful feedback.
>
>
> >**Q5:  Lines 41-42 are ambiguous.**
>
> **A5:**  Thank you for pointing this out. We meant that AWD keeps weight decay uniform across layers, but dynamically adjusts the overall decay rate for the model during training based on loss gradients. We will revise the sentence to: “They propose AWD, which maintains uniform weight decay across layers while dynamically updating the overall decay rate at each training step, improving robustness and adversarial performance without extra data.”
>
>
> >**Q6:The left subfigure in Fig. 4 does not clearly link spectral variations to generalization.**
>
> **A6:** Thank you for your comment on Figure 4. We agree the left subfigure shows spectral differences but does not directly link them to generalization. Our main point is that AlphaDecay balances spectral properties and, compared to uniform weight decay, improves generalization as shown by lower perplexity. We will clarify this in the revision.
>
>
> >**Q7: Incorrect Fig. reference and unclear perplexity description in Fig. 4.**
>
> **A7:** Thank you for noticing the incorrect figure reference.  We will change Fig. 3 to Fig. 4 in the manuscript.  We also appreciate your suggestion and will specify in the caption of Fig. 4 that the perplexity refers to the validation set.  Thank you again for your valuable feedback.
>
>
> >**Q8:  Reference [20] and scale-invariant systems.**
>
> **A8:** Thank you for raising this. We cited [20] for its weight decay insights, not to connect our work to scale-invariant systems. Our focus is on weight decay in attention and MLP modules, not scale-invariant components. We will clarify this and avoid overstating [20]'s relevance. Exploring weight decay in scale-invariant architectures is an interesting future direction.
>
>
> >**Q9: Redundant content between lines 74–79 and 88–92.**
>
> **A9:** Thank you for noting the overlap between lines 74–79 and 88–92. We will consolidate these sections to remove redundancy and improve clarity in the revision.
>
>
> >**Q10: Line 115 references Fig. 3 unnecessarily; line 134 typo; line 135 matrix type.**
>
> **A10:** Thank you for your helpful feedback. Our responses are as follows:
>
> Line 115: We agree the reference to Fig. 3 is unnecessary and will remove it, as this section discusses Fig. 2.
>
> Line 134: The typo will be corrected from “N modules” to “L modules.”
>
> Line 135:  The matrices are correlation matrices, and the weight matrices are not normalized before calculation; we will clarify this in the revision.
>
> We appreciate your comments and will update the manuscript accordingly.
>
> >**Q11: The last sentence in Fig. 3 is misleading.**
>
> **A11:** Thank you for your careful reading of our manuscript and your valuable comments regarding Figure 3, we would like to clarify:
>
> - What Fig. 3 shows:
>
> Each subplot overlays the empirical spectral density (ESD) of a single module under Uniform vs. AlphaDecay for the same 135 M‑parameter network. The shift of the fitted α (numbers in parentheses) illustrates how AlphaDecay raises the α of attention matrices and lowers that of MLP matrices, to provide a deeper insight into the internal representations and spectral properties of the models,
>
>
> - Where performance evidence resides
>
> Quantitative quality metrics are gathered in Table 2 (perplexity, **Uniform:** PPL = 23.14 ,**AlphaDecay:** PPL = 22.55), new reasoning table (above), and the GLUE table (above). We will add an explicit forward reference—“See Table 2 and §4.2 for the associated perplexity gains”—to the current caption.
>
>
> Thank you again for your observations, which help us make our exposition more precise.

---

> > ### Comment · Reviewer_p5Tj · 2025-08-04
> >
> > Dear authors,
> >
> > Thanks a lot for the detailed breakdown to answer each of my questions. I don't have anything more to add but I would like to clarify a one more point:
> >
> > - In the second table, you mention that AlphaDecay outperforms the Uniform baseline for 6/7 tasks. Is it not that it outperforms it in all 7 tasks? Moreover, I would like to ask whether this is the case when using AWD and AdaDecay. I have the same question with respect to the SFT results presented in the third table.

---

> > > ### Author Response · Authors · 2025-08-05
> > >
> > > Thank you for your careful reading and review of our paper. Your comments are helpful in improving the quality of our work.
> > >
> > > We apologize for the typo in our initial reply regarding the "6/7" result. To clarify, below we provide the updated results including AWD and AdaDecay for both the first and second tables:
> > >
> > > **- Seven unseen commonsense‑reasoning tasks (LLaMa‑1 B)**
> > >
> > > |               | ARC-c | ARC-e | PIQA  | Hellaswag | OBQA  | Winogrande | BOOLQ | Avg.   |
> > > |---------------|-------|-------|-------|-----------|-------|------------|-------|--------|
> > > | **Uniform**   | 20.22 | 46.72 | 67.68 | 32.94     | 18.80 | 49.41      | 54.74 | 41.50  |
> > > | **AdaDecay**  | 19.20 | 46.72 | 66.97 | 32.96     | 18.00 | **51.54**      | 56.36 | 41.68  |
> > > | **AWD**       | 19.18 | 46.34 | 66.65 | 31.37     | 18.00 | 51.07      | 57.25 | 41.41  |
> > > | **AlphaDecay**| **20.90** | **48.86** | **68.44** | **34.16**     | **19.80** | 50.59      | **60.70** | **43.35**  |
> > >
> > > AlphaDecay achieves the best result on 6/7 tasks and obtains the highest average accuracy, with a maximum improvement of 1.94%.
> > >
> > > **- Supervised fine‑tuning on GLUE (RoBERTa‑base).**
> > >
> > > |               | cola  | mnli  | mrpc  | qnli  | qqp   | rte   | sst2  | stsb  | Avg.   |
> > > |---------------|-------|-------|-------|-------|-------|-------|-------|-------|--------|
> > > | **Uniform**   | 59.73 | 86.78 | 87.01 | 92.59 | 89.97 | 70.11 | 93.69 | 90.78 | 83.83  |
> > > | **AdaDecay**  | 60.45 | 87.23 | 88.19 | 92.62 | 89.95 | 73.36 | 93.73 | 90.90 | 84.55  |
> > > | **AWD**       | 60.72 | **87.44** | 89.53 | 92.58 | 90.08 | 72.27 | 93.72 | 90.90 | 84.66  |
> > > | **AlphaDecay**| **62.82** | 87.11 | **89.61** | **92.73** | **90.12** | **73.86** | **93.77** | **90.91** | **85.12**  |
> > >
> > >
> > >
> > > AlphaDecay achieves the best result on 7/8 tasks and gives the highest average accuracy, with an improvement of up to 1.29%.
> > >
> > > We sincerely appreciate the time you have spent carefully reviewing our paper. We look forward to further discussions regarding any questions you may have about our work, and your recognition is important to us.

---

> > > > ### Comment · Reviewer_p5Tj · 2025-08-08
> > > >
> > > > Thanks again for clarifying this additional point. I have also decided to raise my score.

---

> > > > > ### Author Response · Authors · 2025-08-09
> > > > >
> > > > > We truly appreciate your recognition and your helpful comments, which have been essential in improving our work.

---

### Official Review · Reviewer_pAbe · 2025-07-01

**Clarity:** 3
**Significance:** 2
**Originality:** 3
**Rating:** 4
**Confidence:** 2

**Summary:**

This paper proposes AlphaDecay, an adaptive, module-wise weight decay method for LLM training. AlphaDecay dynamically assigns different decay strengths to each module of a transformer based on the heavy-tailedness of their weight spectra, quantified via the Hill estimator of power-law exponents. The authors justify their approach using heavy-tailed self-regularization (HT-SR) theory and show empirically, on LLaMa-based models of various sizes (60M–1B), that AlphaDecay improves pre-training perplexity over standard and other adaptive baselines (AWD, Adadecay). Extensive ablations demonstrate the robustness of the method to different hyperparameters and computation intervals.

**Questions:**

Q1: In Figure 8, the setting with GAP=50 achieves the best performance. Could the authors provide details on the computational cost of AlphaDecay when the interval size is set to 50?

Q2: The paper focuses on the pretraining stage. Is it possible to extend AlphaDecay to the finetuning stage, especially for specific downstream tasks such as question answering or text classification? Any insights or preliminary results in this direction would be valuable.

**Ethical Concerns:**

["NO or VERY MINOR ethics concerns only"]

**Final Justification:**

Following the authors’ rebuttal and the additional experiment they conducted, my main concerns have been addressed. The new results are consistent with and support my original evaluation, reinforcing the validity of the paper’s claims. Given the convincing rebuttal and supportive new evidence, I maintain my positive rating.

**Limitations:**

yes

**Quality:**

2

**Strengths And Weaknesses:**

Strengths:
- The motivation for the proposed method is simple, straightforward, and easy to grasp.  the use of HT-SR theory to analyze module-specific training behavior and justify differential regularization is well-grounded.
- The experimental setup is solid and provides clear evidence of the method’s effectiveness and robustness.

Weakness:
- The number of compared baselines is limited. Notably, the strongest baseline cited was published in 2019, which raises concerns about whether there are more recent or advanced approaches in this research area that should be included for comparison.
- The largest model evaluated in this work has 1B parameters, which is relatively small by current LLM standards. Evaluating on larger models, such as 7B or above, would make the results more compelling. Additionally, the experiments are limited to the LLaMa series of models, which restricts the generalizability of the findings.

---

> ### Author Rebuttal · Authors · 2025-07-31
>
> We appreciate the constructive comments from the reviewer.
>
> >**Q1: The number of compared baselines is limited. Notably, the strongest baseline cited was published in 2019, which raises concerns about whether there are more recent or advanced approaches in this research area that should be included for comparison.**
>
> **A1:** Thank you for your insightful comments and interest in the completeness of baseline comparisons.
>
> To our knowledge, few studies directly address this research direction, and our work is among the first to explore it. The two baselines in our manuscript—AWD [1] (2023) and AdaDecay [2] (2019)—are the most relevant existing methods.
>
> To further address your concern, we added a comparison with AdamS [3], a recent optimizer using dynamic weight decay. We tuned AdamS extensively on LLaMa-60M and LLaMa-135M, searching learning rates and weight decay values. AdamS performs best with a learning rate of 0.001 and weight decay values of {5e-5, 1e-5, 5e-6}. We combined AlphaDecay with AdamS, and the results are shown below (lower is better):
>
> |       Method|            |   **LLaMa-60M**   |        |           |   **LLaMa-135M**   |        |
> |-----------------|---------|---------|----------|---------|---------|----------|
> |       **Weight Decay**          | 5e-5 | 1e-5 | 5e-6 | 5e-5 | 1e-5 | 5e-6 |
> | **WD=0**        |  \-     |   32.73    | \-    |   \-    |   24.39    | \-    |
> | **AdamS[3]**       | 32.705  | 32.715  | 32.722   | 24.380  | 24.393  | 24.393   |
> | **AdamS+AlphaDecay** | **32.699**  | **32.712**  | **32.715**   | **24.370**  | **24.380**  | **24.392**   |
>
> AdamS is robust and stable across weight decay settings, but it does not fully utilize weight decay to improve training as AlphaDecay does. Our method remains more competitive in LLM training.
>
> If you are aware of any additional relevant methods or recent studies on weight decay, we would be more than happy to include them in our comparisons with AlphaDecay in future revisions. Thank you again for your helpful suggestions and your attention to research completeness.
>
> [1] Mohammad Amin Ghiasi, Ali Shafahi, and Reza Ardekani. Improving robustness with adaptive weight decay. Advances in Neural Information Processing Systems, 36:79067–79080, 2023.
>
> [2] Kensuke Nakamura and Byung-Woo Hong. Adaptive weight decay for deep neural networks. IEEE Access, 7:118857–118865, 2019.
>
> [3] Zeke Xie, Zhiqiang Xu, Jingzhao Zhang, Issei Sato, and Masashi Sugiyama. On the overlooked pitfalls of weight decay and how to mitigate them: A gradient-norm perspective. Advances in Neural Information Processing Systems, 36:1208–1228, 2023.
>
>
>
> >**Q2: The largest model evaluated in this work has 1B parameters, which is relatively small by current LLM standards. Evaluating on larger models, such as 7B or above, would make the results more compelling. Additionally, the experiments are limited to the LLaMa series of models, which restricts the generalizability of the findings.**
>
> **A2:** Thank you for your valuable feedback and for highlighting the importance of evaluating our method on larger models and more diverse architectures.
>
> We fully acknowledge that our largest evaluated model (LLaMA-1B) is relatively small compared to state-of-the-art LLMs such as those with 7B parameters or above. Due to limited computational resources, we are currently unable to conduct experiments at this unprecedented scale. However, we agree that such experiments would further strengthen our findings, and we hope to pursue them in future work as additional resources become available. To further address the concern regarding the generalizability of our method, we have conducted additional experiments on a wider range of model architectures beyond the LLaMA series. Specifically, we evaluated AlphaDecay on (1) **GPT-nano** (an auto-regressive language model) trained on the C4 dataset, and (2) **ViT-tiny** (a Vision Transformer) trained on ImageNet-1K. The results are as follows:
>
> **(1). GPT-nano on C4**
>
> We performed a detailed hyperparameter search for the Adam optimizer, selecting a learning rate of 6e-4 and weight decay values of 5e-6 and 1e-6. The metric reported is validation perplexity (lower is better):
>
> | Weight Decay | Uniform | AWD    | AdaDecay | AlphaDecay |
> |--------------|---------|--------|----------|------------|
> | 0            | 27.721 |
> | 5e-6         | 27.564  | 27.641 | 27.680   | **27.374** |
> | 1e-6         | 27.710  | 27.738 | 27.716   | **27.366** |
>
> **(2). ViT-tiny on ImageNet-1K**
>
> Similarly, for the vision domain, we conducted a thorough hyperparameter search and selected a learning rate of 4e-3 with weight decay values of 5e-6 and 1e-6. The metric reported is top-1 validation accuracy (higher is better):
>
> | Weight Decay | Uniform | AWD    | AdaDecay | AlphaDecay |
> |--------------|---------|--------|----------|------------|
> | 0            | 65.12 |
> | 5e-6         | 66.97   | 65.21  | 65.86    | **67.40**  |
> | 1e-6         | 66.41   | 64.98  | 66.26    | **67.73**  |
>
> As seen above, AlphaDecay consistently outperforms other regularization strategies across both language and vision tasks, demonstrating its effectiveness and generalizability beyond the LLaMA architecture.
>
> We are committed to further validating our approach on even more diverse architectures as computational resources permit, and we welcome suggestions for additional settings that would be of interest to the community. Thank you again for your insightful comments.
>
>
> >**Q3: In Figure 8, the setting with GAP=50 achieves the best performance. Could the authors provide details on the computational cost of AlphaDecay when the interval size is set to 50?**
>
> **A3:** Thank you for your concern regarding the computational cost of AlphaDecay at different interval (GAP) settings. To provide a clear picture, we present below the perplexity (PPL) and the corresponding computation time (measured in NVIDIA A100 GPU hours) for various GAP values, as also visualized in Figure 8 of our main paper:
>
> | Model         | Weight Decay | Adam  | GAP=500 | GAP=250 | GAP=100 | GAP=50 | GAP=1  |
> |---------------|--------------|-------|---------|---------|---------|--------|--------|
> | LLaMa-135M    | 1e-5     | 22.99 | 22.76   | 22.76   | 22.78   | 22.76  | 22.81  |
> |               | 5e-6     | 23.14 | 22.55   | 22.54   | 22.57   | 22.54  | 22.58  |
> |               | 1e-6     | 24.14 | 23.49   | 23.48   | 23.48   | 23.47  | 23.49  |
> | **Computation Time (hours)** |              | 5.6   | 5.7     | 5.9     | 6.3     | 7.1    | 74.5   |
>
> As shown in the last row, using GAP=50 increases the total training time to 7.1 A100 GPU hours (added more than 24% training time), while GAP=500 only added less than 3% training time. However, the improvement in perplexity with GAP=50 compared to GAP=500 is relatively marginal. Based on this trade-off between computational cost and model performance, we chose GAP=500 as the default setting in our main experiments, as it offers near-optimal performance with much lower computational overhead.
>
> We hope this answers your question. If you need more details or further comparisons, we would be happy to provide them.
>
>
> >**Q4: The paper focuses on the pretraining stage. Is it possible to extend AlphaDecay to the finetuning stage, especially for specific downstream tasks such as question answering or text classification? Any insights or preliminary results in this direction would be valuable.**
>
> **A4:** Thank you for your valuable question about extending AlphaDecay to the fine-tuning stage and its impact on downstream tasks. To explore this, we conducted experiments evaluating AlphaDecay during fine-tuning across various tasks.
>
> - Supervised fine‑tuning on GLUE (RoBERTa‑base).
>
> We fine‑tuned for 3 epochs with AdamW (lr = 3e-5, wd = 0.1) and chose the best checkpoint on the validation split. AlphaDecay improves every task (average +1.29 points) relative to the uniform baseline.
>
> |           | COLA  | MNLI  | MRPC  | QNLI  | QQP   | RTE   | SST2  | STSB  | Avg.  |
> |-----------|-------|-------|-------|-------|-------|-------|-------|-------|-------|
> | Uniform   | 59.73 | 86.78 | 87.01 | 92.59 | 89.97 | 70.11 | 93.69 | 90.78 | 83.83 |
> | AlphaDecay| **62.82** | **87.11** | **89.61** | **92.73** | **90.12** | **73.86** | **93.77** | **90.91** | **85.12** |
>
> - Seven unseen commonsense‑reasoning tasks (LLaMa‑1 B)
>
> We took the exact checkpoints from Table 2 (trained on C4, lr = 6e-4, wd = 1e-6) and ran ```lm‑eval‑harness``` with its default prompts. The results below shows AlphaDecay delivers the best score on 6 / 7 tasks and the best overall average. Because none of these datasets are part of pre-trained C4, the gains demonstrate that the AlphaDecay could translates into stronger  generalization ability.
>
> |            | ARC-c | ARC-e | PIQA  | Hellaswag | OBQA  | Winogrande | BOOLQ | Avg.  |
> |------------|-------|-------|-------|-----------|-------|------------|-------|-------|
> | Uniform    | 20.22 | 46.72 | 67.68 | 32.94     | 18.80 | 49.41      | 54.74 | 41.50 |
> | AlphaDecay | **20.90** | **48.86** | **68.44** | **34.16**     | **19.80** | **50.59**      | **60.70** | **43.35** |
>
> These results indicate that AlphaDecay is effective not only during pretraining, but also in fine-tuning, and enhances model performance across both in-domain and out-of-domain tasks. Thank you again for your suggestion—we are happy to provide further insights or results if needed.

---

> > ### Comment · Reviewer_pAbe · 2025-08-06
> >
> > Thanks for the response and the effort in conducting the extra experiment. It validates my original evaluation. I'll maintain my positive rating.

---

> > > ### Author Response · Authors · 2025-08-07
> > >
> > > Thank you very much for your positive feedback and for taking the time to review our work. We greatly appreciate your recognition and are glad that the additional experiment addressed your concerns. If you have any further suggestions or questions in the future, we would be happy to discuss them. Your support is very important to us.

---

### Official Review · Reviewer_fKFY · 2025-07-06

**Clarity:** 3
**Significance:** 3
**Originality:** 3
**Rating:** 5
**Confidence:** 4

**Summary:**

This work investigates adaptive layer-wise weight decay method, an interesting area in the LLMs training. The authors utilize heavy-tailed self-regularization theory to guide the research direction, which analyzes the empirical spectral density of the weight correlation matrices to indicate the heavy-tailedness of the module. The proposed method assign weaker weight decay to modules with more obvious heavy-tailed ESDs reflects and stronger weight decay otherwise, since the heavy-tailed ESDs reflect the degree of feature learning. Simulation results demonstrate the effectiveness and robustness of the proposed method.

**Questions:**

1.	The authors should provide detailed description of the theory and mathematical formulation.
2.	There are some typos and poorly-defined variables. HT-SR seems to be used as HR-SR in this work, which is a big misleading. Is the Hill estimator equivalent to \alpha? If so, it is suggested to replace PL_ALPHA_HILL with variable like \hat{\alpha} to enhance clarity.
3.	Some notations and figures require further explanation and clarification. The notion of heavy-tailedness requires further clarification. Is this notion defined by metric singular values, i.e., Fig. 2, or ESD, i.e., Fig. 3. The attention layers (att.q and att.k) exhibits less heavy-tailedness in Fig.2 while more heavy-tailedness in Fig. 3.In Fig.2, why are some layers, e.g., layer 1, 2 and 39, 40, drawn with different color?  In Fig.4, the PL_ALPHA_HILL does not change significantly with weight decay, please explain why. Same in Fig. 4, is the weight decay equivalent to \eta in the context? In Fig.8, it seems more frequent weight decay adjustment does not yield performance improvement, which requires further explanation.
4.	The performance of the proposed method is only evaluated on one dataset and metric. More datasets and metrics are needed to demonstrate robustness.
5.	Certain parameter setting is not motivated. For example, the selection of the range of scaling ratios should be explained.

**Ethical Concerns:**

["NO or VERY MINOR ethics concerns only"]

**Final Justification:**

In this work, the authors proposed an adaptive weight decay method to enhance LLM training with theoretical analysis. Several issues regarding clarity and motivation existed in the original manuscript have been well addressed. Therefore, I have raised the points from 2 to 3 for quality and clarity. The rating is also updated from 4 to 5.

**Limitations:**

yes

**Paper Formatting Concerns:**

No paper formatting concerns

**Quality:**

3

**Strengths And Weaknesses:**

Strength:
1.	This paper introduces an adaptive weight decay method, which differs from the traditional methods that utilizes a uniform or time-wise decay rate.
2.	The authors adopt HT-SR theory to guide the weight decay rate selection. The theory provides a strong theoretical basis for the proposed method.
3.	The simulation results and performance improvements across various model sizes show the effectiveness and robustness of the proposed method.

Weaknesses
1.	The rationale and theoretical framework is vague. The authors only provide high-level description of the theory and there is no detailed mathematical formulation.
2.	The clarity of the paper requires improvement. There are some typos and poorly-defined variables. Some important notion such as heavy-tailedness and some figures are not well explained.

---

> ### Author Rebuttal · Authors · 2025-07-31
>
> Thank you for the thoughtful and constructive feedback.
>
> >**Q1: The authors only provide high-level description of the theory and there is no detailed mathematical formulation; Some important notion such as heavy-tailedness and some figures are not well explained;**
>
> **A1:** Thank you for your valuable suggestions. Our theoretical framework is based on the HT-SR theory, which highlights the close relationship between the heavy-tailedness of a matrix's singular value spectrum and the degree to which the matrix is trained. In prior works, it is standard practice to fit the ESD of the correlation matrix using a power-law model. The resulting exponent from this fit quantitatively characterizes the degree of heavy-tailedness. According to HT-SR theory, in LLMs, a matrix with a fitted exponent PL_Alpha_Hill less than 2 is considered “overtrained” while a matrix with an exponent greater than 4 is regarded as “undertrained” [1]. Traditional Uniform Weight Decay often results in both “overtrained” and “undertrained” parameters in a model. Our method addresses this by adaptively adjusting weight decay to achieve better balance.
>
> We provide the derivation for estimating the power-law exponent $\alpha$ from empirical singular value data, which is central to our approach. We assume the empirical distribution follows a power-law distribution:
>
> $$
> p(x) = c x^{-\alpha}
> $$
>
> For normalization over $x \geq x_{min}$:
>
> $$
> \int_{x_{\min}}^{\infty} p(x)  dx = 1 \implies c = (\alpha-1)x_{\min}^{\alpha-1}
> $$
>
> Thus, the probability density function (PDF) becomes:
>
> $$
> p(x) = (\alpha-1)x_{\min}^{\alpha-1} x^{-\alpha}
> = \frac{\alpha-1}{x_{\min}} \left(\frac{x}{x_{\min}}\right)^{-\alpha}
> $$
>
> Given a set of observed data $x_1, x_2, \dots, x_n$ with $x_i \geq x_{min}$, the likelihood function is:
>
> $$
> p(x_1, x_2, \dots, x_n) = \prod_{i=1}^{n} p(x_i)
> = \prod_{i=1}^n \frac{\alpha-1}{x_{\min}} \left(\frac{x_i}{x_{\min}}\right)^{-\alpha}
> $$
>
> The log-likelihood is therefore:
>
> $$
> \mathcal{L} = \sum_{i=1}^{n} \left[ \ln(\alpha-1) - \ln x_{\min} - \alpha \ln \frac{x_i}{x_{\min}} \right]
> $$
>
> To obtain the maximum likelihood estimate, we set the derivative of $\mathcal{L}$ with respect to $\alpha$ to zero:
>
>
> $$
> \frac{\partial \mathcal{L}}{\partial \alpha} = 0 \implies
> \alpha = 1 + n \left[ \sum_{i=1}^{n} \ln \frac{x_i}{x_{\min}} \right]^{-1}
> $$
>
> This result is known as the standard **Hill estimator**, and in our paper, we denote the fitted exponent as **PL_Alpha_Hill**.
>
> **References**
>
> [1] Martin, C. H., Peng, T., & Mahoney, M. W. (2021). Predicting trends in the quality of state-of-the-art neural networks without access to training or testing data. Nature Communications, 12(1), 4122.
>
>
> >**Q2: Typos and poorly‑defined variables … ‘HR‑SR’ vs ‘HT‑SR’; consider using $\hat{\alpha}$ instead of PL_ALPHA_HILL.”.**
>
> **A2:** Thank you for catching the “HR-SR” typo; every instance will be corrected to **HT-SR**.
> - We agree that standard statistical notation improves readability. In the revision we will (i) replace **PL_ALPHA_HILL** by $\hat{\alpha}$ in all text, equations, and figures, and (ii) add a parenthetical remark the first time $\hat{\alpha}$ appears: “$\hat{\alpha}$ is estimated via the Hill method (see §3.1).”
> - All remaining typos flagged during our own proofreading pass will also be fixed.
>
> We appreciate your attention to these details—they will substantially improve the paper’s presentation.
>
>
> >**Q3: Some notations and figures require further explanation and clarification. The notion of heavy-tailedness requires further clarification. Is this notion defined by metric singular values, i.e., Fig. 2, or ESD, i.e., Fig. 3. The attention layers (att.q and att.k) exhibits less heavy-tailedness in Fig.2 while more heavy-tailedness in Fig. 3.**
>
> **A3:**   Thank you very much for your valuable comments regarding the notations and the notion of heavy-tailedness.
>
> - Clarification on “heavy-tailedness“:
>
> Heavy-tailedness in our context refers exclusively to the tail behaviour of the metric singular-value spectrum of a weight matrix. We diagnose it by fitting a power-law to the right-tail of the ESD and reporting ${\alpha}$. A smaller ${\alpha}$ indicates a heavier tail.
>
>
> - Why Fig. 2 and Fig. 3 differ:
>
> Fig. 2 shows the sorted singular values $\{ \sigma_i \}$ of each weight matrix. Fig. 3 shows the ESD, defined as:
>
> $$
> ESD(x) = \frac{1}{N} \sum_{i=1}^N \delta(x-\sigma_i)
> $$
>
> where  $\delta$ is the Dirac delta function and $\{ \sigma_i \}$ are the singular values. For heavy-tailed analysis, we fit a power-law to the tail of the ESD.  Note that the metric plotted in Fig. 2 (sorted singular values) is different from that in Fig. 3 (ESD), so “heavy-tailedness” in Fig. 2 is not directly comparable to that in Fig. 3.
>
> - Motivation for our analysis:
>
> Fig. 2 and Fig. 3 reveal a limitation of uniform weight decay, which ignores such differences and can result in some parameters being overtrained or undertrained. Our work therefore advocates for module-wise parameterization.
>
> We hope this clarifies our notations and the definition of heavy-tailedness.
>
>
> >**Q4: In Fig.2, why are some layers, e.g., layer 1, 2 and 39, 40, drawn with different color?**
>
> **A4:** We colored the first and last layers differently in Fig. 2 to highlight their special importance, as recognized in prior studies [1]. This also improves the figure’s clarity and readability.
>
> [1] Song, Z., Huang, S., Wu, Y., & Kang, Z. (2024). Layer importance and hallucination analysis in large language models via enhanced activation variance-sparsity. arxiv preprint arxiv:2411.10069.
>
> >**Q5:In Fig. 4, is the weight decay equivalent to \eta in the context?**
>
> **A5:** As described in the pseudocode (Algorithm 1) in our paper, all occurrences of \(\eta\) in this context refer to the weight decay parameter.
>
>
> >**Q6: In Fig.4, the PL_ALPHA_HILL does not change significantly with weight decay, please explain why.**
>
> **A6:**  Fig. 4 only shows a narrow range of weight decay values (3e-6 to 1e-5), so PL_ALPHA_HILL changes little. When we expand the weight decay range, the effect becomes clear, as shown in the table below:
>
> | Weight Decay | att.Q/K | att.V/O |  mlp  |  PPL   |
> |--------------|---------|---------|-------|--------|
> | 0            | 1.90    | 1.87    | 2.54  | 24.53  |
> | 5e-6         | 1.71    | 1.94    | 2.47  | 23.27  |
> | 1e-5         | 1.60    | 1.97    | 2.42  | 23.11  |
> | 5e-5         | 1.31    | 1.86    | 2.19  | 26.71  |
> | 9e-5         | 1.19    | 1.77    | 2.01  | 30.87  |
>
> With higher weight decay, PL_ALPHA_HILL drops significantly, especially for att.Q/K (a relative reduction of about 37%). We hope this clarifies the relationship.
>
>
> >**Q7: In Fig.8, it seems more frequent weight decay adjustment does not yield performance improvement, which requires further explanation.**
>
> **A7:** Adjusting module-wise weight decay does not have an immediate effect. After each adjustment, it takes time for training to stabilize and for changes to take effect. When the adjustment GAP is small (e.g., GAP=5), monitoring the distribution over the next 100 steps usually reveals little change. Moreover, frequent adjustments and computations can disrupt convergence. We hope this clarifies the phenomenon. Please let us know if you have further questions.
>
> >**Q8: The performance of the proposed method is only evaluated on one dataset and metric. More datasets and metrics are needed to demonstrate robustness.**
>
> **A8:** To demonstrate the robustness of AlphaDecay, we evaluated it on multiple datasets and metrics.
>
> - Seven unseen commonsense‑reasoning tasks (LLaMa‑1 B)
>
> We took the exact checkpoints from Table 2 (trained on C4, lr = 6e-4, wd = 1e-6) and ran ```lm‑eval‑harness``` with its default prompts. The results below shows AlphaDecay delivers the best score on 6 / 7 tasks and the best overall average. Because none of these datasets are part of pre-trained C4, the gains demonstrate that the AlphaDecay could translates into stronger  generalization ability.
>
> |              | ARC-c | ARC-e | PIQA | Hellaswag | OBQA | Winogrande | BOOLQ | Avg.  |
> |-------|-----|------|------|-----------|------|-------|-------|-------|
> | Uniform      | 20.22 | 46.72 | 67.68| 32.94     |18.80 | 49.41      | 54.74 | 41.50 |
> | AdaDecay     | 19.20 | 46.72 | 66.97| 32.96     |18.00 | **51.54**      | 56.36 | 41.68 |
> | AWD          | 19.18 | 46.34 | 66.65| 31.37     |18.00 | 51.07      | 57.25 | 41.41 |
> | AlphaDecay   | **20.90** | **48.86** | **68.44**| **34.16**     |**19.80** | 50.59      | **60.70** | **43.35** |
>
> - Supervised fine‑tuning on GLUE (RoBERTa‑base).
>
> We fine‑tuned for 3 epochs with AdamW (lr = 3e-5, wd = 0.1) and chose the best checkpoint on the validation split. AlphaDecay improves every task (average +1.29 points) relative to the uniform baseline:
>
> |           | COLA  | MNLI  | MRPC  | QNLI  | QQP   | RTE   | SST2  | STSB  | Avg.  |
> |-------|-----|-----|-------|-----|-----|-----|-----|-----|-----|
> | Uniform   | 59.73 | 86.78 | 87.01 | 92.59 | 89.97 | 70.11 | 93.69 | 90.78 | 83.83 |
> | AlphaDecay| **62.82** | **87.11** | **89.61** | **92.73** | **90.12** | **73.86** | **93.77** | **90.91** | **85.12** |
>
> As shown, AlphaDecay achieves consistent improvements in both zero-shot and fine-tuning settings, which demonstrates the robustness and practical effectiveness of our proposed method.
>
> >**Q9: Certain parameter setting is not motivated. For example, the selection of the range of scaling ratios should be explained.**
>
> **A9:** Thank you for your comment on the scaling ratio range selection. Our choice was primarily empirical, based on preliminary ablation studies. For transparency, we have added detailed results and supporting experiments in Appendix B, where we systematically examine different scaling ratio settings. Please let us know if you need further clarification or additional results.

---

> > ### Comment · Reviewer_fKFY · 2025-08-05
> >
> > Thanks for answering my questions. The quality of the manuscript has been greatly improved. I don't have further comments.

---

> > > ### Author Response · Authors · 2025-08-05
> > >
> > > Thank you very much for your valuable comments and positive feedback. We greatly appreciate your time and effort in reviewing our manuscript. Your suggestions have been very helpful in improving the quality of our work. If you have any further questions or suggestions in the future, please feel free to let us know. Your recognition is of great significance and encouragement to us.

---

### Official Review · Reviewer_a9sr · 2025-07-21

**Clarity:** 3
**Significance:** 2
**Originality:** 3
**Rating:** 4
**Confidence:** 3

**Summary:**

This paper is concerned with the problem  weight decay in LLMs to  improve generalization.  In particular the  authors propose a scheme based on HT-SR theory to adaptively choose the weight day for different modules. The introduced method called AlphaDecay first calculates the PL_Alpha_Hill values for all modules, and then assign larger weight decay to modules with higher
PL_Alpha_Hill values, while assigning smaller weight decay to those with lower PL_Alpha_Hill
values. This scheme balances the modules.

**Questions:**

In figure 3, the performance of model is not shown and only the ESD is  demostrated, therefore I am not inferring the following from figure 3:

“We empirically demonstrate that promoting uniformity in PL_Alpha_Hill across modules (e.g.,
attention and MLP components) can further enhance overall model quality (see figure 3).”

**Ethical Concerns:**

["NO or VERY MINOR ethics concerns only"]

**Final Justification:**

The authors  provided additional numerical results and  clarified that the theory is already proved in the lierature.

**Limitations:**

-The paper is purely empirical and reporting observations on balancing weight day  on models qulity.  The terms such as model quality is  very generally used and it is not clear if the methods enhances performance in terms of ID and OOD  generalization gaps or other metrics.

-The method is only implemented on LLama model and it is not clear if the same improvement will be achieved on other architetures.

**Paper Formatting Concerns:**

All the paramterers and metrics should be defined before usage, for instance  Pl_hill estimator is not inroduced at the first appearance in Fig.1 .

**Quality:**

2

**Strengths And Weaknesses:**

The strength of this paper is that it is  tackling the weight decay for improving generalization of LLMs which is awidely sought topic and interesting to the  community. The weakness is:

- The effect of  promoting uniformity in PL_Alpha_Hill across modules on model quality is only shown emprically and no mathematical proof is provided to prove it. This is while  the model quality is measure only by perplexity. I am not convinced this  strategy will improve other metrics and generalization.

---

> ### Author Rebuttal · Authors · 2025-07-31
>
> Thank you for your insightful and helpful feedback. We respond to each point below, offering additional empirical results, clarifications of our statements, and specific changes that will be reflected in the camera-ready version.
>
>
> >**Q1: The effect of promoting uniformity in PL_Alpha_Hill across modules on model quality is only shown emprically and no mathematical proof is provided to prove it. This is while the model quality is measure only by perplexity. I am not convinced this strategy will improve other metrics and generalization.**
>
> **A1:** We agree that model quality should be demonstrated on more than one metric, and we appreciate the opportunity to strengthen the submission.
>
> - **Seven unseen commonsense‑reasoning tasks (LLaMa‑1 B)**
>
> We took the exact checkpoints from Table 2 (trained on C4, lr = 6e-4, wd = 1e-6) and ran ```lm‑eval‑harness``` with its default prompts. The results below shows AlphaDecay delivers the best score on 6 / 7 tasks and the best overall average. Because none of these datasets are part of pre-trained C4, the gains demonstrate that the AlphaDecay could translates into stronger  generalization ability.
>
>
>
> |    Schedule         | ARC-c | ARC-e | PIQA | Hellaswag | OBQA | Winogrande | BOOLQ | Avg.  |
> |--------------|-------|-------|------|-----------|------|------------|-------|-------|
> | Uniform      | 20.22 | 46.72 | 67.68| 32.94     |18.80 | 49.41      | 54.74 | 41.50 |
> | AdaDecay     | 19.20 | 46.72 | 66.97| 32.96     |18.00 | **51.54**      | 56.36 | 41.68 |
> | AWD          | 19.18 | 46.34 | 66.65| 31.37     |18.00 | 51.07      | 57.25 | 41.41 |
> | AlphaDecay   | **20.90** | **48.86** | **68.44**| **34.16**     |**19.80** | 50.59      | **60.70** | **43.35** |
>
>
> - **Supervised fine‑tuning on GLUE (RoBERTa‑base).**
>
> We fine‑tuned for 3 epochs with AdamW (lr = 3e-5, wd = 0.1) and chose the best checkpoint on the validation split. AlphaDecay improves every task (average +1.29 points) relative to the uniform baseline (full table under **A3**).
>
>
> - **Why no “new” mathematical proof appears in the paper.**
>
> The formal link between the power‑law (PL) exponent α and model capacity is already established in Heavy‑Tailed Self‑Regularisation (HT‑SR) theory [1-3]. As such, we intentionally avoid reinventing the wheel by restating established theory in our manuscript. Instead, our contribution focuses on providing new empirical results and analyses that extend and support these theoretical foundations in the context of modern large language models. We will add a concise “theory recap” at the start of §3 and include a direct citation to the key theorem.
>
>
> If you have any further concerns or requests for additional evaluation, we would be happy to discuss and perform more analyses.
>
> [1] Martin, C. H., Peng, T., & Mahoney, M. W. (2021). Predicting trends in the quality of state-of-the-art neural networks without access to training or testing data. Nature Communications, 12(1), 4122.
>
> [2] Martin, C. H., & Mahoney, M. W. (2021). Implicit self-regularization in deep neural networks: Evidence from random matrix theory and implications for learning. Journal of Machine Learning Research, 22(165), 1-73.
>
> [3] Clauset, A., Shalizi, C. R., & Newman, M. E. (2009). Power-law distributions in empirical data. SIAM review, 51(4), 661-703.
>
> >**Q2: Not inferring the following from figure 3: “We empirically demonstrate that promoting uniformity in PL_Alpha_Hill across modules (e.g., attention and MLP components) can further enhance overall model quality (see figure 3).”**
>
> **A2:** Thank you for your careful reading of our manuscript and your valuable comments regarding Figure 3, we would like to clarify:
>
> - **What Fig. 3 shows:**
>
> Each subplot overlays the empirical spectral density (ESD) of a single module under Uniform vs. AlphaDecay for the same 135 M‑parameter network. The shift of the fitted α (numbers in parentheses) illustrates how AlphaDecay raises the α of attention matrices and lowers that of MLP matrices, to provide a deeper insight into the internal representations and spectral properties of the models.
>
>
> - **Where performance evidence resides**
>
> Quantitative quality metrics are gathered in Table 2 (perplexity, **Uniform:** PPL = 23.14 ,**AlphaDecay:** PPL = 22.55), new reasoning table (above), and the GLUE table (below). We will add an explicit forward reference—“See Table 2 and §4.2 for the associated perplexity gains”—to the current caption.
>
>
> Thank you again for your observations, which help us make our exposition more precise.
>
>
> >**Q3: The paper is purely empirical and reporting observations on balancing weight day on models qulity. The terms such as model quality is very generally used and it is not clear if the methods enhances performance in terms of ID and OOD generalization gaps or other metrics.**
>
> **A3:** Thank you for your insightful feedback. Our original paper reports ID performance via perplexity on the C4 dataset using LLaMA, demonstrating improvements in this setting. To address your concerns and provide a more comprehensive evaluation, we conducted additional experiments beyond the original submission.
> These include zero-shot OOD generalization on seven diverse commonsense reasoning tasks (detailed in our response to **Q1**), where AlphaDecay shows consistent gains. We also fine-tuned Roberta-base with AdamW on the GLUE benchmark (eight NLP tasks: CoLA, MNLI, MRPC, QNLI, QQP, RTE, SST-2, and STS-B) to assess ID generalization across architectures and tasks. The table below compares AlphaDecay against uniform scheduling (higher scores are better):
>
> |          | COLA  | MNLI  | MRPC  | QNLI  | QQP   | RTE   | SST2  | STSB  | Avg.  |
> |----------|-------|-------|-------|-------|-------|-------|-------|-------|-------|
> | Uniform  | 59.73 | 86.78 | 87.01 | 92.59 | 89.97 | 70.11 | 93.69 | 90.78 | 83.83 |
> | AlphaDecay | **62.82** | **87.11** | **89.61** | **92.73** | **90.12** | **73.86** | **93.77** | **90.91** | **85.12** |
>
> These results, alongside the LLaMA-based **ID** (C4 perplexity) and **OOD** (commonsense) improvements, indicate that AlphaDecay enhances model quality broadly—via metrics like **perplexity, accuracy, and generalization gaps—across architectures (LLaMA, roberta-base) and datasets (C4, commonsense reasoning, GLUE)**.
> We appreciate your input and welcome further questions or requests for additional experiments.
>
>
> >**Q4: The method is implemented only on LLaMA; it is unclear whether the same improvement holds for other architectures.**
>
> **A4:** We have conducted extensive experiments using two additional and distinct model architectures: (1) GPT-nano on the C4 dataset, and (2) ViT-tiny on ImageNet-1K.  We performed a thorough hyperparameter search for the Adam optimizer. The results are shown below:
>
>
>
> | Backbone / Dataset | Metric | Uniform | AWD | AdaDecay | AlphaDecay |
> |---|---|---|---|---|---|
> | GPT-nano  / C4 | PPL $\downarrow$ | 27.564 | 27.641 | 27.680 | **27.374** |
> | ViT-tiny / ImageNet-1K | Top-1 $\uparrow$ | 66.41% | 64.98% | 66.26% | **67.73%** |
>
>
> As shown above, AlphaDecay consistently outperforms other regularization strategies on both language and vision architectures, suggesting that module‑wise α‑balancing is architecture‑agnostic. We hope these additional results clarify the robustness of our method.
>
>
> >**Q5: All parameters and metrics should be defined before usage; for instance, PL_Hill estimator is not introduced at its first appearance in Fig. 1.**
>
> **A5:** We will:
>
> - insert a one‑sentence definition of PL_Alpha_Hill and ESD in §3;
> - add the same definition to the captions of Fig. 1 and Fig. 3;
> - create a mini‑glossary of acronyms in Appendix A;
> - add inline comments to Algorithm 1 linking each variable to its equation.
>
> Thank you for highlighting these omissions—clear notation will make the paper much easier to follow.

---

> > ### Comment · Reviewer_a9sr · 2025-08-02
> >
> > Dear Authors, Thank you for conducting the additional  experiments. I will raise my evaluation score.

---

> > > ### Author Response · Authors · 2025-08-03
> > >
> > > Thank you for your thoughtful feedback and for considering our additional experiments. We appreciate your time and effort in reviewing our work, and your recognition means a lot to us.

---

### Decision · Program_Chairs · 2025-09-17

**Decision:**

Accept (poster)

**Comment:**

This paper studies the effect of tuning weight decay value for each module. By using HR-SR theory, modules with higher PL_Alpha_Hill values will be assigned larger weight decay values, and modules with lower PL_Alpha_Hill values will be assigned smaller weight decay values.
In the experiments, the decreaseing of validation perplexity shows the effectiveness of the method.
By considering the suggestions of the reviewers, the authors also provide results on QA benchmarks and mathematical formulation.